# Assessing the Vulnerability of Water Balance to Climate Change at River Basin Scale in Humid Tropics: Implications for a Sustainable Water Future

**Kashish Sadhwani** [1] and **T. I. Eldho** [1,2,*]

1 Department of Civil Engineering, Indian Institute of Technology Bombay, Mumbai 400076, India; kashishksh@gmail.com
2 Interdisciplinary Program in Climate Studies, Indian Institute of Technology Bombay, Mumbai 400076, India
* Correspondence: eldho@civil.iitb.ac.in

**Abstract:** Sustainability in hydrology aims at maintaining a high likelihood of meeting future water demands without compromising hydrologic, environmental, or physical integrity. Therefore, understanding the local-scale impact of global climate change on hydrology and water balance is crucial. This study focuses on assessing the impact of climate change on water balance components (precipitation, surface runoff, groundwater flow, percolation, etc.) at the river basin scale in a humid tropical region. The Periyar river basin (PRB) in Kerala in India is considered as a case study and the SWAT hydrological model is adopted to obtain the water balance components. Three general circulation models are considered under two shared socioeconomic pathways (SSP 245 and SSP 585) emission scenarios assess the impact of climate change until 2100. For the PRB, the results demonstrate a significant increase in streamflow (>65%) and runoff (>40%) in the mid (2041–2070) and far (2071–2100) future under both the SSP scenarios, indicating a potential vulnerability to future floods. Conversely, in the near future under SSP 585, a decrease in runoff (−15%) and nominal changes in streamflow (−5%) are observed. Spatially, the eastern sub-basins and the west coast of the Periyar river basin are projected to experience higher precipitation events, while the central region faces reduced precipitation and low flow rates. The findings emphasize the need for proactive and sustainable management of water resources, considering irrigation requirements, groundwater discharge, and flood control measures, to mitigate the negative effects of climate change and prevent water stress/surplus situations in specific sub-basins. This study enhances our understanding of climate change impacts on water balance and emphasizes the significance of sustainable water resource management for an effective response. By integrating scientific knowledge into policy and management decisions, we can strive towards a resilient water future within a changing climate.

**Keywords:** climate change; CMIP6; GCM; humid tropics; hydrology; SWAT; water balance; Western Ghats

## 1. Introduction

The availability of clean water is crucial to public health, environmental functioning, and economic growth. The security of our water supplies is threatened by rising population, fast urbanization, shifting diets as countries grow, excessive abstraction, and worsening pollution (United Nations World Water Development Report 2018) [1]. In addition, climate change poses a serious threat to water supplies, economic growth, and political stability due to the potential for negative feedback loops (IPCC 2022) [2]. Temperature rises, changes in precipitation pattern, delayed monsoons, and increased frequencies of floods and droughts have made the situation more vulnerable (IPCC 2012) [3].

Climate change can have indirect impacts on water balance by affecting the health of ecosystems and the services they provide. For instance, changes in temperature and precipitation patterns could alter the distribution and abundance of plants and animals,

which could in turn affect the functioning of wetlands, forests, and other ecosystems that play important roles in regulating the water cycle. The rise in temperatures, which can lead to increased evaporation and transpiration, and changes in precipitation patterns may result in less water being available for surface runoff and the recharge of groundwater. As a result, there may be less water available for irrigation, drinking, and other uses. Contrarily, heavy rainfall events becoming more common as a result of climate change, as well as changes in land use and land cover, can exacerbate the risk of flooding. Floods can cause significant damage to infrastructure, disrupt transportation and communication networks, and lead to loss of life and property.

Land use changes, and natural and anthropogenic transformations, are likewise responsible for substantial influence on watershed hydrology [4,5]. However, climate change has been identified to be dominant over land use changes [6,7] in affecting the hydrology and water balance, although the results vary across regions [8–10]. Thus, there is a need to better understand the effect of climate change on water balance components at river basin scale for sustainable water resource management. Sustainability aims at maintaining a high likelihood of meeting future demands without compromising hydrologic, environmental, or physical integrity. With the competing priorities of different stakeholders involved, this task becomes more challenging.

Hydrological models are notably useful for determining the regional hydrologic implications of changes in temperature, precipitation, and other climatic factors. Hydrologic models are mainly classified as empirical models, conceptual models, and physically based models. Due to the inclusion of parameters with physical interpretations, physically based hydrological models are preferred over other model types for assessing the effects of climate change [11]. Another advantage is that they can be used in many scenarios and provide a lot of information about the role of the parameters involved. Water balance models are particularly appealing for water resource studies of climatic changes due to their ability to incorporate monthly or seasonal variations in climate, snowfall and snowmelt algorithms, soil moisture, groundwater, and natural climatic variability [12]. With proper calibration and validation, these models can replicate near-true conditions. The Soil and Water Assessment Tool (SWAT) is one such physically based hydrological model that has been identified to accurately predict annual and monthly streamflow in a variety of environmental conditions [13,14]. These findings make the model a useful instrument for hydrological estimates of water attributes on a watershed scale.

General Circulation Models (GCMs) are widely recognized as a highly effective means of examining the physical mechanism of the earth's surface and atmosphere system, and of providing reliable insights about past, present, and future climate [8,15]. Representative Concentration Pathways (RCPs) are climate change scenarios developed as part of the Coupled Model Intercomparison Project Phase 5 (CMIP5) that may portray a wide range of potential future climatic situations. Due to this, RCP scenarios are an appealing possible strategy for further study in emissions mitigation and impact assessments [16]. However, with the recent advancement in climate science, a new set of experiments have been conducted under CMIP6 (Phase 6) [2]. CMIP6 experiments consider more complex earth system models and microphysical processes in addition to the carbon cycle component (major highlight in CMIP5). This includes better understanding of ocean heat uptake, sea level rises, responses to volcanic forcings, feedback from aerosol and atmospheric chemistry, and sea level rises from land ice sheets [17]. Additionally, the shared socioeconomic pathways (SSP) scenarios of CMIP6 consider socioeconomic factors like future population and economic growth combined with climate change adaptation and mitigation efforts; it represents better realizations of future scenarios [18].

The spatial resolution of the GCMs is very coarse for use with hydrological models to simulate basin-scale or sub-grid hydrological processes, hence the downscaling method is used to bring it a finer resolution [8,15,19,20]. Ghosh and Mujumdar (2007) [19] stated that GCM simulation results might be highly uncertain due to an unknown future and inadequate knowledge of globally changing geophysical processes. Tebaldi et al. (2005) [21]

identified the same and quantified the uncertainty associated with the regional climate models. Thus, in such cases, it is desirable to adopt a multi-model ensemble as it optimizes forecast efficacy and minimizes uncertainty [21,22].

Humid tropical regions have been a point of attraction for human settlements due to their unique ecohydrology [23,24]. The Western Ghats, a humid tropical region in India, is one such region that has been experiencing various changes in water resources due to climate change. This includes changes in precipitation patterns with an increasing trend in the southern part of the region, whereas there is a decreasing trend in the upper part of the region [25]. This contrasting trend in southwest monsoon rainfall in the northern and southern Western Ghats has been reported in other studies, as well [26]. The weakening of vertical velocity and reduced summer mean rainfall over the orographic region has also been reported [27]. Other than this, certain regional studies have reported this area to be vulnerable to water scarcity [28], whereas in some regions there has been an increase in flooding events [29].

Considering the higher vulnerabilities in the humid tropics, this study examines the effects of climate change on water balance at the river basin scale to gain insights into the sustainability issues related to water resources. The Periyar river basin, a humid tropical watershed in the Western Ghats region, is selected to demonstrate the methodology of this study. The Periyar river basin is a complex watershed with several anthropogenic constraints. Climate change impact assessment based on the latest CMIP6 scenarios creates insights into water resource management for this area. Thus, assessing the impact of future climate change and considering latest SSP scenarios in relation to water balance components within this complex watershed is the main focus of this study. This area has not been previously addressed in any other study in this region. The SWAT model is employed to conduct this assessment, allowing for a comprehensive evaluation of the effects of climate change. The datasets from three GCMs are utilized to analyze the change until 2100 under the SSP 245 and SSP 585 scenarios.

## 2. Materials and Methods

### 2.1. Study Area Description and Input Data Details

2.1.1. Study Area Description

The Periyar river is the second-longest river (>244 km in length) in Kerala, India, with a catchment area of approximately 4793 km$^2$. The basin has an inverted 'L' shape, and its overall drainage pattern is dendritic in nature. It is a west-flowing river that originates at an elevation of 2438 m above mean sea level (AMSL) in the Western Ghats mountain range and drains into the Arabian Sea. The highest elevation in the basin is at 2695 m (AMSL), and the lowest is near the Arabian Sea (Figure 1c). The watershed lies between latitude 9°16′ N to 10°20′ N and longitude 76° E to 77°30′ E (Figure 1a). The PRB has a tropical-humid climate, with rainfall concentrated over the months of June through November. The mean annual rainfall in the PRB is 3200 mm (CWC, 2018) [30], and the maximum and minimum temperature range from 25 °C to 32 °C and 14 °C to 19 °C, respectively, in the basin [31]. Since it is a large-elongated basin, the average annual temperature varies from 28 °C near upstream to 30 °C near the downstream end. The average annual evapotranspiration of the basin is approximately 850 mm. The PRB is majorly covered with plantation (52.02%) and forests (33.31%), and the soil texture mainly consists of clay and loam soil. The forest area mainly consists of tropical evergreen trees and plantations, predominantly featuring rubber, eucalyptus, teakwood, coconut, and areca nut. The agro-climatic conditions of the region are favorable for the cultivation of cash crops, including rice, millet, coffee, pepper, and cardamom, which are the main source of income for the area. The Periyar river, being a perennial river, is a vital source of water in the central parts of Kerala, serving a population of more than 4,391,362 people (Census 2011, https://censusindia.gov.in/, accessed on 1 May 2023). There are three major reservoirs: Mullaperiyar Dam (capacity: 443.23 × 10$^6$ m$^3$), Idukki Dam (capacity: 5550 × 10$^6$ m$^3$), Idamalayar Dam (capacity: 1089 × 10$^6$ m$^3$) (https://www.kseb.in/, ac-

cessed on 15 February 2019), [32] and one hydrological observation station at Neeleshwaram (10°12′ N 76°5′ E) in the basin (Figure 1a). The average annual runoff from Neeleshwaram gauging station is recorded as $6686 \times 10^6$ m$^3$ [31]. The maximum daily discharge measured between 1989 and 2017 peaked at 6324 m$^3$/s. Notably, the last week of July or the first and second weeks of August experience the majority, or around 80%, of the daily discharge at over 2000 m$^3$/s. The dams serve the purpose of electricity generation, flood control, and fulfilling irrigation water demands in the region. The Mullaperiyar Dam serves the purpose of diverting water to the eastern side, in the rain shadow region, whereas the Idamalayar and Idukki Dams' major purpose is to generate electricity for the region. The PRB is a complex watershed experiencing various constraints related to water distribution and reservoir operations. For instance, a major portion of the water stored in the Idukki reservoir is channeled outside of the river basin. Furthermore, a certain portion of the area comes under forest reserves, which makes the area limited by several anthropogenic constraints. The details are discussed by Sadhwani et al. (2023) [33].

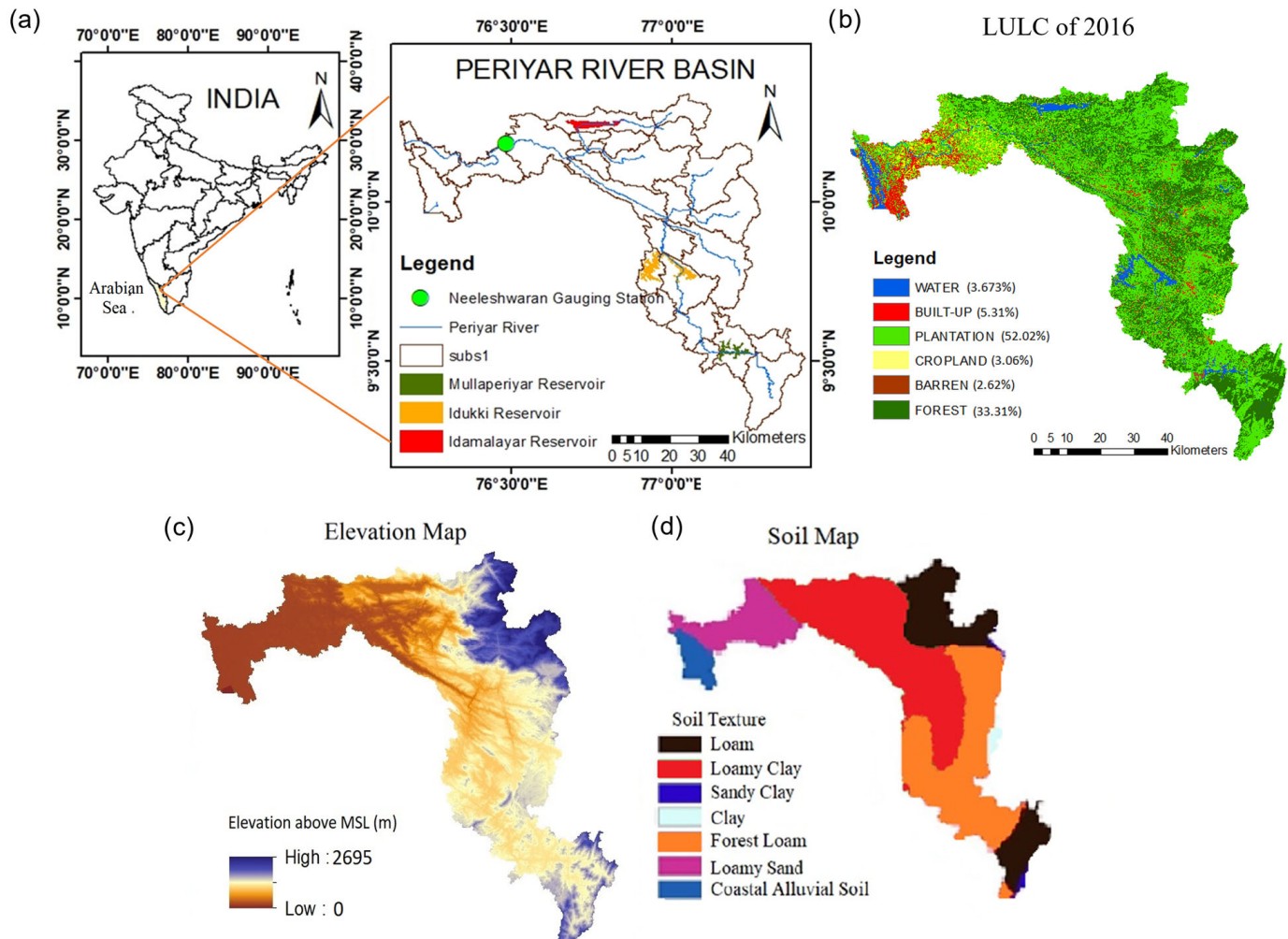

**Figure 1.** (**a**) Periyar river basin and sub-basins, (**b**) LULC (2016), (**c**) elevation map, and (**d**) soil texture of Periyar river basin.

### 2.1.2. Topographical and Meteorological Data

The topography data used in the study includes slope, elevation, flow direction, and flow accumulation. These were produced from a CartoDEM digital elevation model (DEM), procured from the National Remote Sensing Centre (NRSC) with the ISRO having a 30 m grid resolution (Figure 1c). The land use land cover (LULC) of the PRB for the year 2016 is mostly plantations (52.02%), followed by forests (33.31%), built-up land (5.21%), water

(3.673%), cropland (3.06%), and barren land (2.62%), as shown in Figure 1b. This land cover distribution was prepared from 2016 LandSAT satellite images (www.earthexplorer.usgs/, accessed on 12 January 2020) using a supervised maximum likelihood classification technique [34]. The classification accuracy and Kappa coefficient were 92% and 0.87, respectively. As per the National Bureau of Soil Survey and Land Use Planning (NBSS & LUP) soil classification, the soil types in the PRB are loamy clay, forest loam, loamy sand, sandy clay, clay, and loam (Figure 1d).

The meteorological datasets used in this study include precipitation, minimum and maximum temperature, solar radiation, and wind speed. The precipitation [35] (0.25° resolution) and temperature [36] (1.0° resolution) were obtained from the India Meteorological Department (IMD) in gridded format.

The wind speed (0.5° resolution) and solar radiation (0.5° resolution) were obtained from the Climate Forecast System Reanalysis (CFSR) data (https://rda.ucar.edu/, accessed on 1 February 2020). All of these datasets were obtained at daily time-step and were linearly interpolated to 0.25° spatial resolution to be used as an input for the SWAT model. To calibrate the SWAT model, gauge discharge data was obtained from the Central Water Commission (CWC), India. The details of the datasets used in the study are mentioned in Table 1.

**Table 1.** Details of the datasets.

| Data Type | Time Period | Resolution Post-Processing | Source |
|---|---|---|---|
| Meteorological Data Historical Precipitation Temperature | 1980–2019 1980–2019 | 0.25° × 0.25° 1.0° × 1.0° | India Meteorological Department (IMD) |
| Solar Radiation Wind Speed | 1980–2019 1980–2019 | 0.5° × 0.5° 0.5° × 0.5° | Climate Forecast System Reanalysis (CFSR) data https://rda.ucar.edu/. |
| Future Precipitation Temperature Wind Speed | 2015–2100 2015–2100 2015–2100 | 0.25° × 0.25° 0.5° × 0.5° 0.5° × 0.5° | GCM (CanESM5, CNRM-CM6-1, and MPI-ESM1-2-LR) |
| Digital Elevation Model | 2005 | 30 m × 30 m | CartoDEM (https://bhuvan.nrsc.gov.in/) |
| Soil Texture | 2012 | 30 arc second | National Bureau of Soil Survey and Land Use Planning (NBSS & LUP) |
| LULC | 2016 | 30 m × 30 m | LandSAT (https://earthexplorer.usgs.gov/) |
| Gauge Discharge | 2000–2015 | Daily | Central Water Commission (CWC) |

### 2.1.3. GCM Climate Data

The future meteorological data inputs, including precipitation, minimum and maximum temperature, and wind speed, are obtained from three GCMs of CMIP6. The GCMs used are National Centre for Meteorological Research (CNRM-CM6) [37], Canadian Earth System Model (CanESM5) [38], and Max Planck Institute for Meteorology Earth System Model (MPI-ESM1-2-LR) [39]. These GCMs are selected on the basis of their ability to replicate the Indian Summer Monsoon [40] and performance in past studies [41]. All of the datasets are obtained in gridded format at different spatial resolutions and then statistically downscaled [40] and bias corrected using a quantile mapping technique [42]. In this study, two socioeconomic pathways, SSP 245 and SSP 585, are considered for analysis from each GCM. The SSP 245 corresponds to the development pathways consistent with historical

patterns and with medium radiating forcings (up to 4.5 W/m$^2$) in 2100. This scenario represents the medium range of the possible future. Conversely, the SSP 585 corresponds to development pathways representing high industrial/economic growth and fossil fuel resource consumption, with minimal efforts to reduce environmental concerns. This scenario corresponds to high radiating forcings (up to 8.5 W/m$^2$) in 2100 [18], with potentially high challenges to mitigation strategies.

### 2.2. SWAT Model Description

The Soil and Water Assessment Tool (SWAT) is a semi-distributed physical hydrological model developed by the U.S. Department of Agriculture (USDA) [43]. SWAT input requires physical parameters, including soil type, LULC, DEM, and meteorological variables, including minimum and maximum temperature, solar radiation, precipitation, and wind speed. SWAT is an effective model for analyzing and predicting the behavior of hydrological changes in water balance components in a watershed [44]. It works on the principle of the water balance equation (Equation (1)) and includes all the major components of the hydrological cycle in a region.

$$W_m = W_0 + \sum_{i=1}^{n} \left( I_{day} - Q_r - ET_0 - w_s - Q_{gw} \right) \tag{1}$$

where $W_m$ is the final water content on the ith day, $W_0$ is the initial water content, $I_{day}$ is the precipitation amount (mm), $Q_r$ is the surface runoff (m$^3$/s), $ET_0$ is evapotranspiration, $w_s$ is the water flow from soil into the vadose zone (mm), and $Q_{gw}$ is the return flow generated from groundwater (m$^3$/s) for n days.

The watershed is segmented into smaller sub-basins utilizing data obtained from the digital elevation model (DEM). These sub-watersheds are then partitioned into uniform groupings based on characteristics such as LULC, soil type, and slope. This grouping process results in the creation of hydrologic response units (HRUs) [45]. The SCS Curve number method (USDA Soil Conservation Service, 1972) is employed to compute the runoff, while channel routing is accomplished through the variable storage method.

The evapotranspiration calculation is based on the Penman–Monteith equation [45]. For groundwater flow, the SWAT model assumes a regional flow pattern for groundwater movement where the water is transmitted from one aquifer to another and, ultimately, to the stream. This transfer of water is assumed by a simple transfer of flow rates.

### 2.3. Methodology

The future meteorological data inputs, including precipitation and minimum and maximum temperature, are obtained from GCMs after statistical downscaling using a non-parametric kernel regression model [46] and then bias corrected, with respect to historical IMD data, using a quantile delta mapping technique [42]. The details of the process applied is discussed in detail by Salvi et al. (2013) [42]. This meteorological input, along with DEM, LULC, and soil texture is used to run the SWAT model. Following calibration, the SWAT model is used to examine the impact of climate change on water balance components. The relative change in the future is enumerated into three time segments: S1 (2021–2040), S2 (2041–2070), and S3 (2071–2100). Finally, an equal weighted ensemble average of the GCM results is enumerated for all three time segments. Figure 2 illustrates the implemented methodological framework.

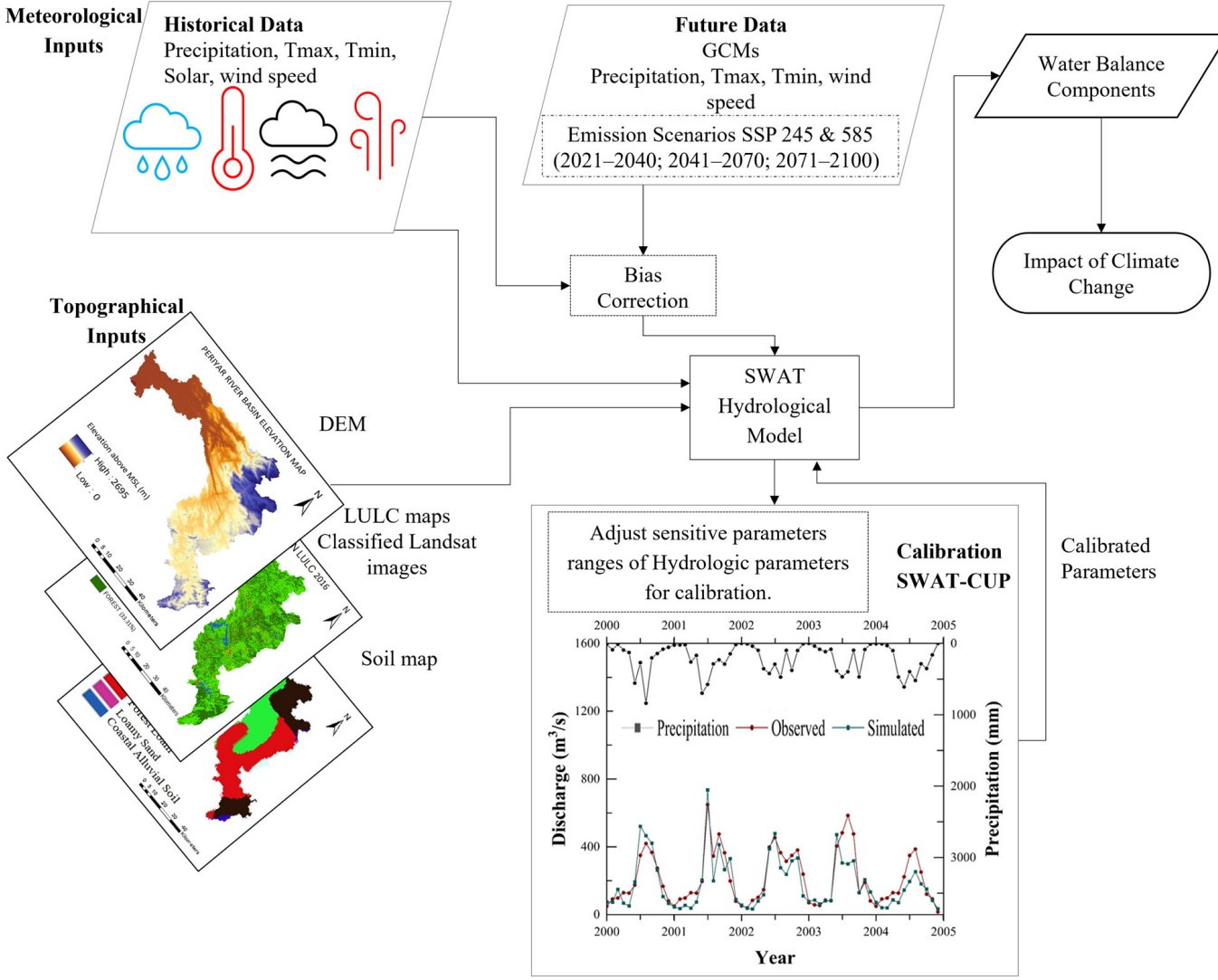

**Figure 2.** Flow chart showing the proposed methodology for assessing the vulnerability of water balance components to climate change using SWAT model.

## 3. Results

### 3.1. GCM Data Assessment

To assess the suitability of GCM data, historical GCM data was compared to IMD precipitation and temperature data from the years 1989 to 2019. The statistics of the observed and GCM-simulated climate variables (and their ensemble average) were compared, and climatology is presented in Figure 3 with the Taylor diagram in Figure 4. In the model cluster, there was a good correlation between the observed and GCM variables. For precipitation, the correlation (r) was greater than 0.85 (Figure 4a, likewise for minimum temperature (correlation > 0.96, Figure 4b) and maximum temperature (correlation > 0.9, Figure 4c) for all GCMs. The bias-corrected variables adequately represented the climatic conditions of the PRB (Figure 3), suggesting its suitability to be used as an input in the SWAT model for future climatic variables using the three GCM datasets based on the SSP scenarios.

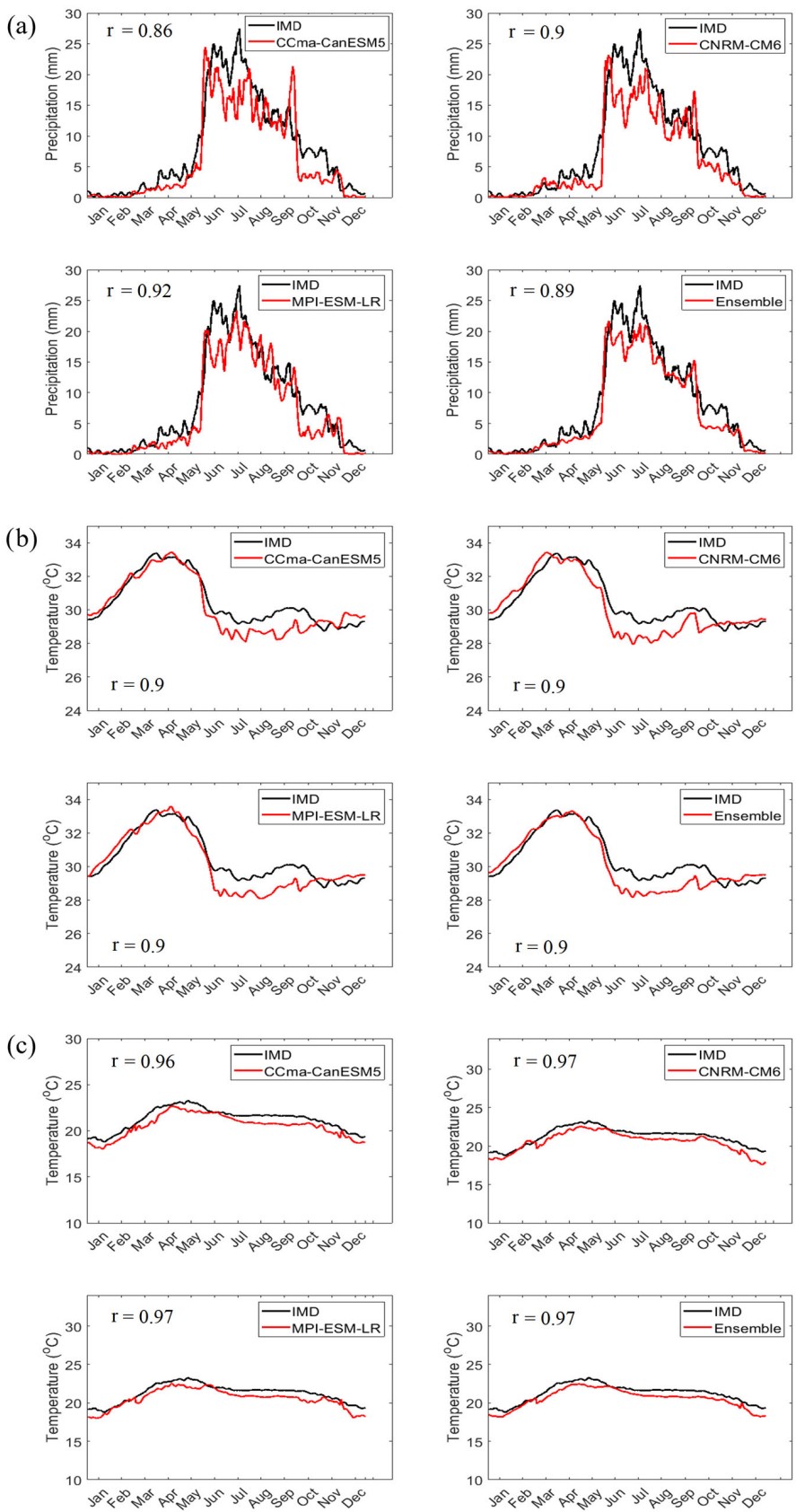

**Figure 3.** Comparison of climatology of GCMs and their ensemble average, with respect to observed climatology (IMD) from 1989–2019, for daily: (**a**) rainfall, (**b**) maximum temperature, and (**c**) minimum temperature. The correlation coefficient (r) value is mentioned in the corner of each figure.

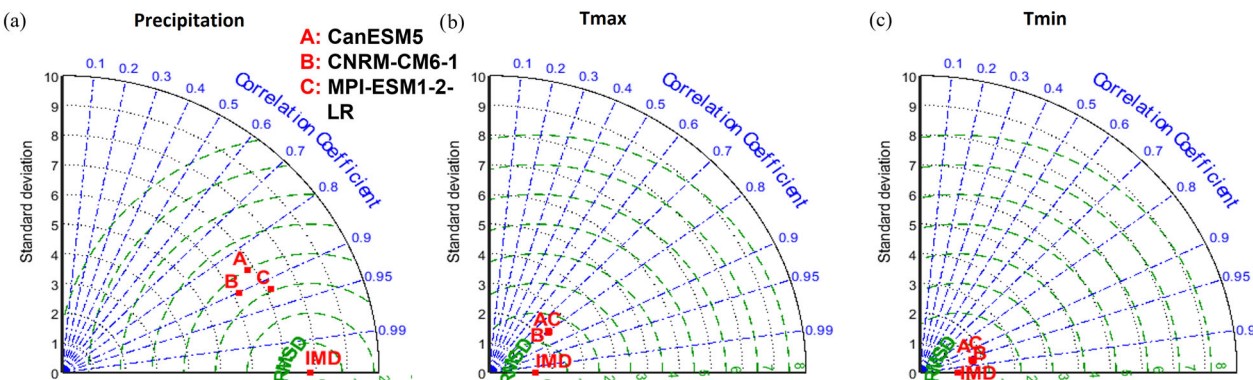

**Figure 4.** Taylor diagram comparing performance of each GCM with respect to the observed data for the historical period (1989–2019): (**a**) rainfall (mm), (**b**) maximum temperature (Tmax) (°C), and (**c**) minimum temperature (Tmin) (°C).

Figure 5 presents the time series for the ensemble of future climatic variables for SSP conditions. The future GCM data reveals an increasing trend in precipitation and maximum and minimum temperature in the PRB (Figure 5a–c). The projected average annual precipitation for the future is expected to range between 2000 mm to 3900 mm for both SSP 245 and SSP 585 scenarios (Figure 5a). Between 2030 and 2070, the precipitation projections for SSP 245 are greater than SSP 585, whereas for the subsequent years SSP 585 shows a higher magnitude than SSP 245. For maximum and minimum temperature, both the SSP 585 and SSP 245 scenarios showed an increasing trend, with the former being higher in magnitude. The increase in maximum and minimum temperatures is up to 2 °C and 4 °C for SSP 245 and SSP 585, respectively (Figure 5b,c).

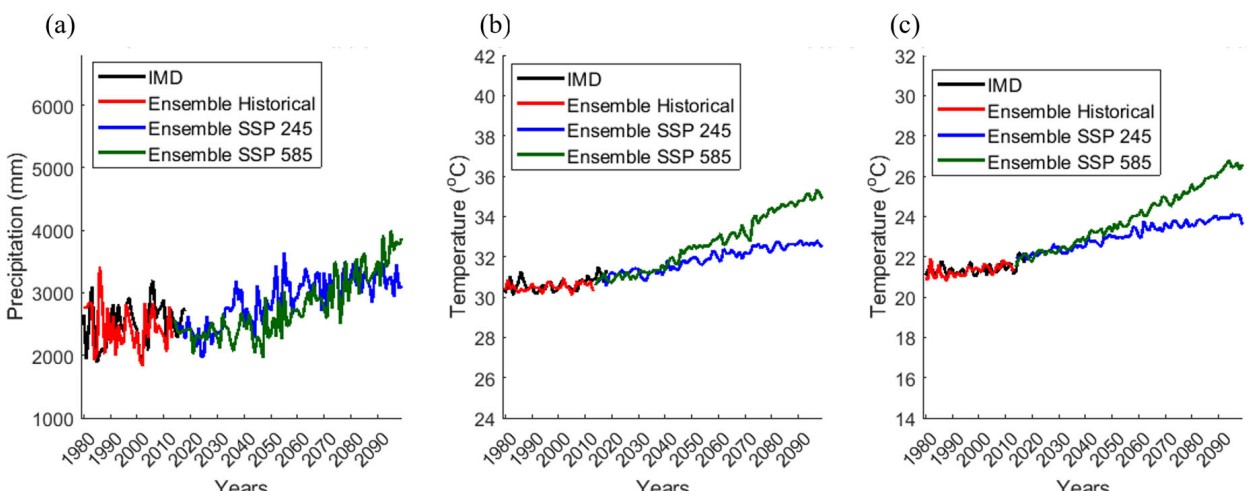

**Figure 5.** Historical (1980–2020) and future (2021–2100) time series of GCM ensemble for the average annual (**a**) precipitation (mm), (**b**) Tmax (°C), and (**c**) Tmin (°C), along with IMD (observed: 1980–2020) values to observe trend.

### 3.2. Calibration and Validation

The SWAT model was set up for the basin, and the watershed was divided into 27 sub-basins with a total area of 4793 sq. km (as depicted in Figure 1a). Monthly streamflow was used to calibrate the model in accordance with the Neeleshwaram gauging station managed by CWC (as shown in Figure 1a). The calibration parameters were adopted from the results of Sadhwani et al. 2022 [5]. The calibrated parameters and their fitted values were CN2 (SCS-CN II value = 68.75, averaged), alpha_bnk (bank storage baseflow factor = 0.063), sol_awc (available soil layer water capacity for plant uptake = 0.274), ESCO (soil evaporation compensation factors = 0.967), alpha_bf (base flow recession constant = 0.0049),

gw_delay (groundwater delay = 125.04), surlag (surface runoff lag time = 19.06), and gw_revap (groundwater revap coefficient for percolation = 0.029). The study performed calibration of the model for the period 2000–2004 (Figure 6a) using the coefficient of determination ($R^2$), Nash-Sutcliffe efficiency (NSE), and Percent Bias (PBIAS) as calibration criteria. The calibration process yielded $R^2$, NSE, and PBIAS values of 0.91, 0.83, and 6.5%, respectively. To validate the model, it was tested for the period 2006–2010 (Figure 6b), and the results showed $R^2$, NSE, and PBIAS values of 0.84, 0.67, and 11.6%, respectively. These results support the suitability of the model for analysis of future climate change impacts.

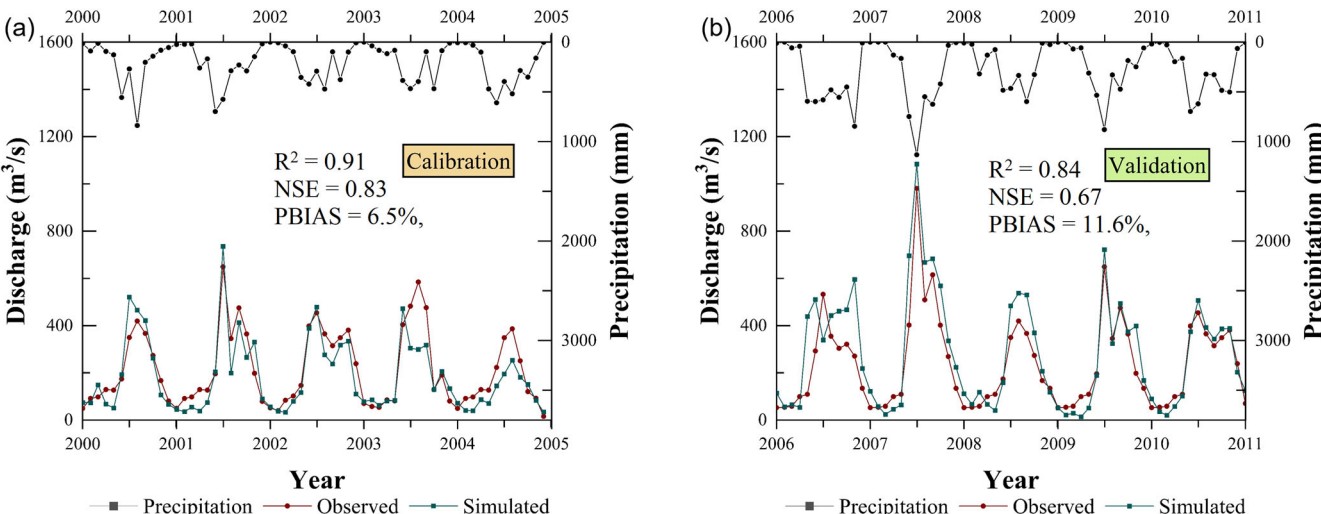

**Figure 6.** Observed and simulated discharge for (**a**) calibration (2000–2004) and (**b**) validation (2006–2010) periods at Neeleshwaram gauging station for the PRB.

### 3.3. Impact of Climate Change on Water Balance Components

Figure 7 shows the spatial distribution of mean annual precipitation, evapotranspiration (ET), surface runoff, and groundwater discharge averaged over the period 1989–2019. The precipitation varies from 700 mm/year to 3400 mm/year across the basin, with higher rainfall near the coast (Westward end) and least rainfall at the eastern end of the basin (Figure 7a). For evapotranspiration (ET), the range varies between 420 mm/year to 1260 mm/year. Higher ET is observed in regions with water bodies and forest cover (Figures 1b and 7b). Regions with crop cover show relatively fewer ET values. This can be attributed to the fact that crops generally have a lower leaf area index and a lesser ability to extract water from the root zone. This results in less water availability for transpiration, leading to lower ET values in regions with crop cover. Surface runoff is also affected by land cover type and the same has been observed in Figure 7c. Other than this, precipitation plays a major role in high surface runoff generation. The maximum surface runoff in the PRB was 2160 mm/year. This is the region witnessing the highest rainfall across the entire basin (Figure 7a,c). Furthermore, the areas with less forest cover generate more runoff (Figures 1c and 7c). The groundwater discharge ranges between 13 mm/year and 1037 mm/year. Groundwater discharge, in this context, refers to the natural flow of water from an aquifer into nearby surface water bodies. It is presumed that there is no extraction of groundwater from the aquifer, as drawing water from an aquifer can upset the equilibrium between groundwater recharge, discharge, and the water stored below the surface. The regions experiencing high rainfall with low elevation terrain show high groundwater recharge, which in turn leads to higher groundwater discharge into the streams. The maximum groundwater discharge is observed near the downstream end of the basin (Figure 7d). This can be related to the above discussion relating to precipitation and elevation. Furthermore, a shallow groundwater level also results in the creation of more pressure pushing water out of the aquifer, which can result in faster groundwater discharge rates.

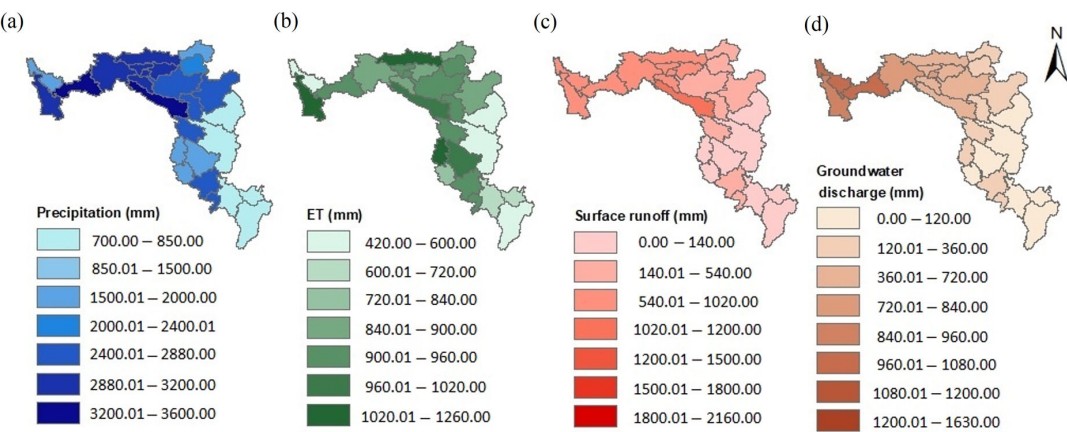

**Figure 7.** Average annual (**a**) precipitation (mm), (**b**) evapotranspiration (mm), (**c**) surface runoff, and (**d**) groundwater discharge (mm) for the historical period (1989–2019) in the PRB.

To analyze the impact of climate change on the water balance components (precipitation, ET, runoff, and groundwater discharge), the spatial change in these characteristics for the three future time segments, S1 (2021–2040), S2 (2041–2070), and S3 (2071–2100), were enumerated, and results are presented in Figures 8 and 9. Furthermore, an overall change in all of the water balance components averaged over the basin is represented in Table 2.

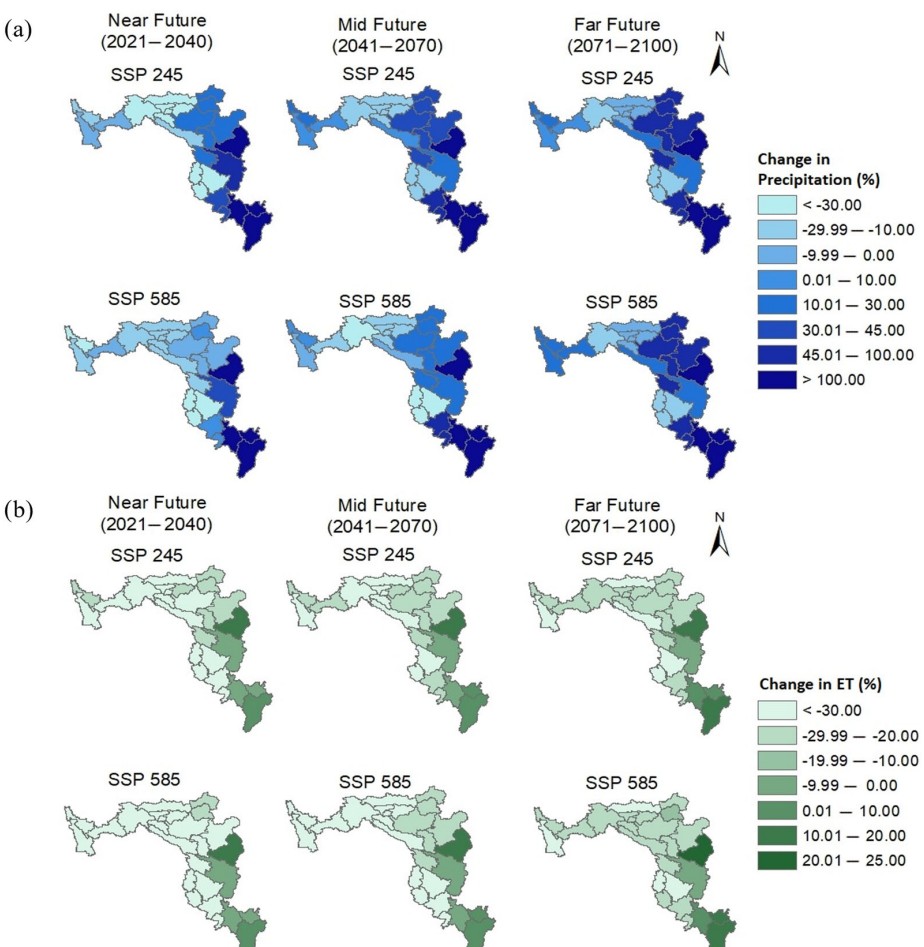

**Figure 8.** Spatial change in average annual (**a**) precipitation (%) and (**b**) evapotranspiration (%) for near (2021–2040), mid (2041–2070), and far (2071–2100) future under SSP 245 and SSP 585 scenarios in the PRB.

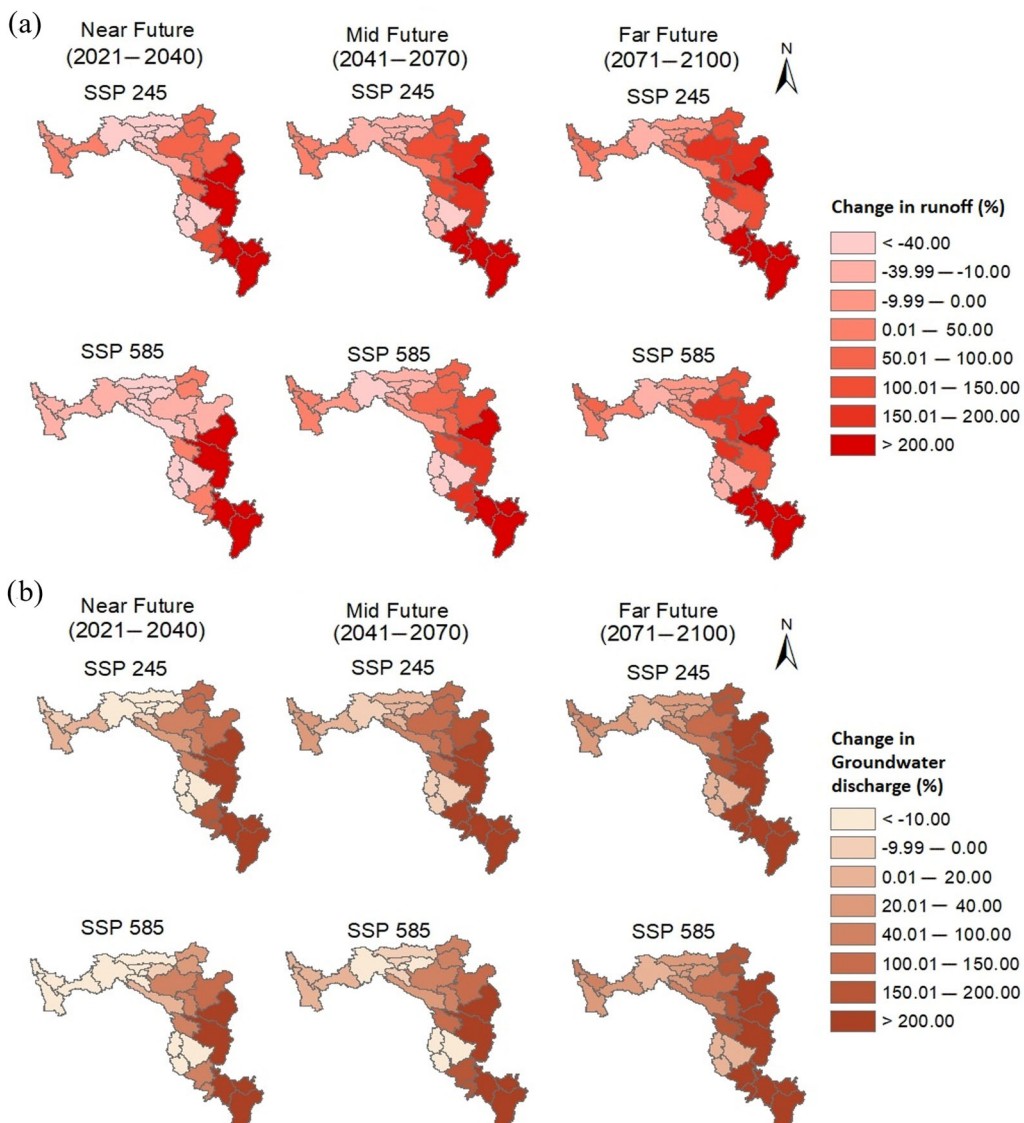

**Figure 9.** Spatial change in average annual (**a**) surface runoff (%) and (**b**) groundwater discharge (%) for near (2021–2040), mid (2041–2070), and far (2071–2100) future under SSP 245 and SSP 585 scenarios in the PRB.

Figure 8a represents the spatial change in precipitation across the PRB for S1, S2, and S3 under SSP 245 and SSP 585 scenarios. The results suggest a reduction in precipitation in the central part of the basin up to −30% in S1 in SSP 245. However, this reduction decreases in S2 and S3, ranging up to −10%. A similar trend is observed in the precipitation of the central part in SSP 585. Near the coast (western side of the basin), the precipitation shows a decrement of approximately −13% in S1. However, in S2 and S3 the precipitation is expected to increase up to 14% and 30%, respectively, in the SSP 245 scenario. For SSP 585, a similar trend is observed with a change in precipitation near the coast of −34%, 1%, and 26% in S1, S2, and S3, respectively. The maximum increase in precipitation (>100%) is observed near the eastern side of the PRB. The increasing trend observed is similar to the other parts of the basin. The reason for such a high magnitude of change in precipitation in this area is the low precipitation values observed in the historical period (Figure 7a). This region is at a high elevation and exposed to various orographic factors influencing the precipitation patterns, which is sometimes not captured well by GCMs.

**Table 2.** Projected change in annual average water balance components due to climate change.

| Water Balance Components | Historical (1989–2019) | % Change | | | | | |
| --- | --- | --- | --- | --- | --- | --- | --- |
| | | SSP 245 | | | SSP 585 | | |
| | | Near (2021–2040) | Mid (2041–2070) | Far (2071–2100) | Near (2021–2040) | Mid (2041–2070) | Far (2071–2100) |
| Precipitation (mm) | 2402.3 | +10.05% | +30.35% | +43.70% | −5.79% | +14.92% | +42.88% |
| ET (mm) | 861.62 | −26.94% | −25.98% | −24.14% | −28.67% | −27.41% | −21.67% |
| Streamflow at Outlet (m³/s) | 156.70 | +56.13% | +94.62% | +118.73% | −5.44% | +65.17% | +113.84% |
| Runoff (mm) | 531.67 | +25.86% | +70.10% | +91.47% | −15.17% | +41.25% | +86.48% |
| Percolation (mm/year) | 530.16 | +32.30% | +58.14% | +78.64% | −0.10% | +43.42% | +76.52% |
| Groundwater Discharge into Stream (mm) | 393.60 | +51.32% | +84.04% | +109.44% | +15.61% | +53.50% | +105.29% |

Figure 8b represents the spatial change in ET across the PRB for S1, S2, and S3. It is observed that the ET decreases in the central and western part of the PRB nearly by −30%, −30%, and −26% in S1, S2, and S3, respectively, for SSP 245. Similar results are observed for the SSP 585 scenario, with a slightly lower magnitude of −32%, −32%, and −23% for S1, S2, and S3, respectively. A higher change in ET is observed on the eastern side of the basin, with +13%, +14%, and +17%, in case of SSP 245, and +10%, +12%, and +21%, in case of SSP 585, for S1, S2, and S3, respectively. The change in ET follows a similar pattern as the change in precipitation, indicating that precipitation is a key factor controlling ET changes. Other than precipitation, land cover also influences the ET rate. Regions with forest cover show a lower magnitude in changes in ET (eastern side of the basin, refer to Figure 1b). Contrastingly, the areas with more urban settlements show higher correlation with precipitation change. This suggests that the influence of precipitation on ET is greater in urban areas.

Other than precipitation and ET, other water balance components that significantly influence the hydrology of a region include surface runoff, groundwater discharge, percolation, and streamflow. These factors uniquely alter the hydrology of a region and demand specific management strategies. The spatial variation in average annual surface runoff and groundwater discharge is represented in Figure 9, and the results for the average annual change of the basin for all of the water balance components is tabulated in Table 2.

The spatial change in average annual surface runoff is represented in Figure 9a. It is observed that the central part of the PRB shows a decrease in runoff by −46%, −20%, and −1% in S1, S2, and S3, respectively, for SSP 245. In contrast, the change is around −44%, −29%, and −5% in the SSP 585 scenario. For the western side, the change is +17%, +25%, and +24% for SSP 245, whereas it is −16%, +10%, and +35% for SSP 585 in S1, S2, and S3, respectively. On the eastern side of the PRB, the change is more than 200% in all the time segments in both scenarios. The change in surface runoff shows an increasing trend, with a spatial pattern similar to precipitation and ET. The immense change in runoff values on the eastern side of the basin can be attributed to two major factors, i.e., the huge difference in precipitation and the high slopes in this sub-region.

Similarly, the spatial change in groundwater discharge is analyzed and the results are shown in Figure 9b. The results suggest a change of −15%, +7%, and +24% for SSP 245 and −35%, −10%, and +20% for SSP 585 in S1, S2, and S3, respectively, in the central region. For the western side, the change is +10%, +22%, and +31% for SSP 254, whereas it is −19%, +1%, and +33% for SSP 585 in S1, S2, and S3, respectively. Similar to precipitation and runoff, the maximum change was observed in the eastern side of the basin, with a change in groundwater flow greater than 200% in both the scenarios and all the time segments.

Furthermore, the results for all of the above-mentioned water balance components, along with percolation and streamflow averaged over the basin, are mentioned in Table 2 and the result for each sub-basin is mentioned in the Supplementary Material (Tables S1–S4). Both percolation and streamflow are expected to increase by +78.64% and +118.73%, respectively, under SSP 245, and +76.52% and +113.84%, respectively, under SSP 585 by 2100. Such changes are alarming and need to be understood better to manage water resources efficiently. It should be noted that here the outlet is considered at the location of Neeleshwaram gauging station (refer to Figure 1a) to analyze the change in streamflow. This site was selected as the calibration for the model was done with respect to this location. Additionally, it should be noted that the change in streamflow is not determined by taking the average over the entire basin because some sub-basins exhibit a positive change while others demonstrate a negative change, which could potentially nullify the overall change in streamflow.

## 4. Discussion

Water resource availability is imperiled by the changing climate, and understanding the impact of climate change on water balance components is critical for ensuring a sustainable future. In this study, the impact of climate change on the water balance components in the Periyar river basin has been discussed. The results show that precipitation is expected to increase in the mid and far future in both SSP 245 and 585 scenarios. In the near future, precipitation increases in the case of SSP 245, whereas it decreases in the SSP 585 scenario. The other water balance components, including streamflow, runoff, and percolation, show an increasing trend in the future. The spatial pattern of changes in the water balance drivers showed a strong correlation among the components. Other than this, the topographical factor plays an important role in this study area. The sub-basins on the eastern side of the watershed have been experiencing high runoff (Figure 9a) in response to the combined effect of high slopes and increased precipitation (Figures 1c and 8a). Contrastingly, the sub-basins in the central part of the river basin, with relatively fewer slopes, experience reduced runoff due to the direct effect of reduced precipitation (Figures 1c and 8a). Contrary to this, the sub-basins at the western end near the coast are expected to experience high runoff with little increase in precipitation in future (Figures 8a and 9a), despite having a flat terrain. This behavior can be related to the land use characteristics of this area, as it has a high coverage of built-up areas (Figure 1b) and even a small amount of precipitation leads to high runoff values in such regions. Thus, land use characteristics play a major role in watershed management. Excess runoff often leads to increased river flow and can become a potential cause of flooding. Thus, it is crucial to give special attention to this aspect of water resource management and flood control. Overall, with precipitation being identified as the main driving component for the other hydrological changes, it is interesting to note that changes in hydrology can be anticipated early. Thus, preparedness against climate change action can be done efficiently and effectively.

For instance, precipitation is the primary driver for runoff, which often leads to floods. The excess runoff drives into river channels and leads to increasing or decreasing streamflow. This can have implications for water management, particularly for activities that rely on a consistent flow of water, such as hydropower generation and irrigation. The runoff can be controlled by several land use management practices, thus reducing the damages caused by floods. Similarly, the groundwater flow is projected to increase in all time periods and under both SSPs, indicating that there may be more water available for use in groundwater systems. This could have implications for water management, particularly for activities that rely on a consistent flow of groundwater, such as agriculture and drinking water supply. Groundwater acts as a reserve water source, especially in case of droughts. Thus, ensuring sustainable management of groundwater systems is the key to dealing with drought-related issues and future water scarcity [47].

An additional aspect of climate change impact on water balance components is the influence on crop productivity and yield. In the PRB, most of the agricultural area is limited to the downstream end (Figure 1b). Projections indicate that in these sub-basins there is a notable decrease in precipitation in near future, whereas there is a nominal decrease in the mid and far future (Figure 8a). The same trend is observed for ET, runoff, and groundwater discharge (Figures 8b and 9a,b). These changing trends raise concerns, as reduced precipitation not only diminishes crop yield but also hampers crop growth. Additionally, runoff plays a crucial role in carrying nutrients from the soil and maintaining soil moisture. Consequently, a decrease in runoff can result in reduced nutrient availability for plant uptake and a decline in soil moisture content. The reduced flow of groundwater is also worrisome, as it can increase the risk of soil salinity and negatively impact soil fertility. However, it is essential to recognize that these changes are multifaceted, requiring further investigation into the associated agricultural risks within the region. Consequently, strategic planning and effective water resource management are imperative to address these challenges and mitigate their potential impacts on agriculture.

This study was conducted in a humid tropical region in India. Humid tropical regions have a unique geography and biodiversity, due to which they have been a point of attraction for human settlements [23]. Considering the high vulnerability of these areas, the results were compared with other similar studies in humid tropical regions. This was done to improve understanding regarding the behavioral change in water balance components that would help in developing a generalized management strategy for water resource management. The results were consistent with studies conducted by Sinha et al. 2020 [48] and Visweshwaran et al. 2022 [9] in the neighboring humid tropical basins in the Western Ghats, where the increase in precipitation led to an increase in streamflow and runoff in the future, using CMIP5 climatic projections. However, regional characteristics play a major role in defining the behavior of water balance components. In the study conducted by Chanapathi et al. 2020 [49] at Krishna river basin, the results were different. There was a reduction observed in rainfall, but the surface runoff and streamflow showed an increase. This suggests the importance of regional studies for understanding the heterogeneity of the different river basins in response to climate change. A study by Nasonova et al. 2021 [50] compared the future change in water balance components in 11 river basins across various parts of the world and highlighted the same issue. Their analysis showed that despite an increase in precipitation and ET in the Amazon river basin, there was no significant change in runoff values [50]. In the case of the Yangtze (sub-tropical climate zone) and Yellow river (temperate climate zone), the increase in precipitation and ET led to a decrease in the runoff [50]. In the case of the Ganges river (humid tropical and sub-tropical climate zone), the surface runoff showed an increment with an increase in precipitation and ET [50]. Such comparative studies shall be promoted to improve understanding related to hydrology sciences. Furthermore, the outcomes of this study were compared with the results of Sadhwani et al. (2023) [33], where the earlier climate models with CMIP5 projection were tested in the Periyar river basin. It was identified that the results were in consensus with the earlier study with runoff, showing a similar spatial change across the basin with little change in magnitude. Thus, this study creates insights about the similarity between the outcomes of CMIP5 and CMIP6 climate models in this study area. Furthermore, it was noticed that the maximum impact of climate change on water balance occurred during the monsoon season (June to September).

Considering the results for the current study area, the projected changes in the water balance components suggest that there may be significant impacts on water availability and management in the future, particularly in areas where water is already scarce. Understanding these changes and their implications can help policymakers and water managers make informed decisions about how to manage and allocate water resources in the future.

## 5. Conclusions

The significance of sustainability in preserving water resources cannot be overstated, particularly given the threat of climate change to the hydrological system and water balance. To ensure effective water resource management, it is crucial to integrate science-based evidence into public policy. Governance strategies for these issues should be adapted to both local conditions and governance frameworks. In such intricate environments, informed decision-making will be critical for developing and implementing effective policies, particularly in the face of climate change.

This study offers valuable insights into how climate change could impact various water balance components at the river basin scale in a humid tropical area, considering the Periyar river basin in South India as a case study. The study utilizes the SWAT hydrological model and different CMIP6 GCMs, under various emission scenarios, to comprehensively understand potential temporal and spatial changes in precipitation, surface runoff, groundwater flow, percolation, and streamflow until 2100. The results show that there is a major increase in streamflow (>65%) and runoff (>40%) in the mid and far future for both of the SSP scenarios. This makes the situation alarming and vulnerable to floods in the future. Therefore, it is crucial to implement targeted measures to address these potential flooding events. However, in case of the near future in SSP 585, the runoff reduces by −15% and the streamflow shows a nominal change of −5%. The spatial variation across the basin shows that the sub-basins in the eastern area and the west coast of the basin will face higher precipitation events, whereas the central region has to deal with less precipitation and low flow rates in future. These findings highlight the main result of the research: demonstrating the spatial variability of responses to climate change across the sub-basins in a large river basin, despite their similar climatic conditions. Overall, precipitation is identified as the governing factor for change in runoff, groundwater discharge, and streamflow in the context of varying climate.

The findings of this study highlight the need for proactive and sustainable management of water resources, including irrigation requirements and groundwater discharge, to mitigate the negative effects of climate change and prevent water stress/surplus situations in the specific sub-basins. This study also underscores the significance of using modeling approaches in developing long-term strategies for water resource management that can support sustainable development in the face of climate change. Some of these strategies may include flood control in areas where excess runoff is observed. Others may include utilizing this information on excess runoff water availability in future to enhance productivity by hydro-power generation, agricultural productivity, and increase water recreation and tourism. Similarly, regions experiencing reduced water availability in the future must be dealt accordingly to utilize water resources more effectively. Furthermore, this study can be considered as a reference for long river catchments with varied topographical changes and reservoir operation constraints in order to deal with climate change issues.

**Supplementary Materials:** The following supporting information can be downloaded at: https://www.mdpi.com/article/10.3390/su15119135/s1, Figure S1: Sub-basin code for Periyar River Basin; Table S1: Projected change in annual average precipitation (mm) due to climate change for each sub-basin; Table S2: Projected change in annual average ET (mm) due to climate change for each sub-basin; Table S3: Projected change in annual average runoff (mm) due to climate change for each sub-basin; and Table S4: Projected change in annual average groundwater discharge (mm) due to climate change for each sub-basin.

**Author Contributions:** Conceptualization, K.S. and T.I.E.; methodology: K.S. and T.I.E.; software, K.S.; validation, K.S.; formal analysis, K.S.; writing and editing, K.S. and T.I.E.; supervision, T.I.E. All authors have read and agreed to the published version of the manuscript.

**Funding:** This research received no external funding.

**Institutional Review Board Statement:** Not applicable.

**Informed Consent Statement:** Not applicable.

**Data Availability Statement:** The datasets analyzed in the study are available from the corresponding author on request.

**Acknowledgments:** The authors would like to express their gratitude to INCCC, the Ministry of Water Resources (now known as the Ministry of Jal Shakti), and the Government of India for supporting the project titled "Impact of Climate Change on Water Resources in River Basins from Tadri to Kanyakumari". Additionally, we would like to acknowledge the Central Water Commission, National Bureau of Soil Survey and Land Use Planning, and Indian Meteorological Department for providing us with valuable hydrological, soil, and meteorological data.

**Conflicts of Interest:** The authors declare no conflict of interest.

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
