# Peer review of "Assessing the Vulnerability of Water Balance to Climate Change at River Basin Scale in Humid Tropics: Implications for a Sustainable Water Future"

_sustainability, doi:10.3390/su15119135_

Round 1

Reviewer 1 Report

Brief summary

This paper, entitled ‘Assessing Vulnerability of Water Balance to Climate Change: Implications for Sustainable Water Future’ this study focuses on assessing the impact of climate change on the water balance components (precipitation, surface runoff, groundwater flow, percolation, etc.) at the river basin scale in a humid tropical region (the Periyar river basin, Kerala, India) using the SWAT hydrological and three general circulation models (GCMs: CanESM5, CNRM-CM6-1, and MPI-ESM1-2-LR ) under two shared socio-economic pathways (SSP 245 and SSP 585) emission scenarios.

The authors concluded that climate change would result in an increase in precipitation and reduction in evapotranspiration, on one hand, and a significant rise in runoff, streamflow, and percolation, on the other hand, through mid (2041–2070) and far (2071–2100) future periods. Such situations often give rise to floods and affect the hydrology. They considered that alteration in the hydrological variables considerably affect the groundwater re charge rate, soil moisture, and crop water requirement, which may create water stress situations. Therefore, they recommended that the developed approaches could assist decision makers in developing strategies for the long-term management of water resources.

The list of literature is very well balanced and updated.

General comments

-       -  Before line 97, you may add a paragraph on the related literature on the specific climate change impacts on the water resources in the study country/region ;

-        - It seems that you based your future projections using the LULC of 2016. Why you haven’t used the LULC projections you developed in your paper published in Water in 2022 ?;

-       -  For the discharge data at the outlet : i) Have you taken into consideration the storage and release from the dams ?; ii) you have data from 2000 to 2015 and in the your above mentioned your paper you have also data from 1991, why did you use for the calibration/validation only the periods : 2000-2004 and 2006-2010 ?

-      -   How you can explain the decrease of projected ET while both temperature and rainfall are increasing ? 

-        - I think speaking of ‘water stress situations’ (line 440) with current and future abundant rain and runoff is not suitable particularly in this humid tropical region. Nevertheless, if you consider that there would be some  specific periods of water stress, I highly encourage you to add a section on scenario analysis : for example adding more dams, digging boreholes, etc.

-        Minor comments

 -    No need of decimals as the study area is large and you are dealing of projections

-         - In fig.1 : a) add : Arabian Sea, Western … Mountaind ; b) what is the difference between Plantation and Cropland ?; c) : inverse the colors (brown for high and blue for low)

Author Response

Assessing Vulnerability of Water Balance to Climate Change in Humid Tropics: Implications for Sustainable Water Future

Kashish Sadhwani and T. I. Eldho

Article number: sustainability-2322136

The authors would like to appreciate the time and effort taken in the review of the manuscript by the Editor and reviewers. We are grateful for the valuable comments and suggestions, which have helped to improve the quality of the manuscript. Further, we thank all the anonymous reviewers for their positive comments. In the revised version, we have attempted to address all the concerns of the Editor and the reviewers.

Please note that the review comments are given in red font while the responses to these comments are given in blue font with “manuscript changes highlighted in Italics within quotes”.

[Line numbers mentioned in the responses are w.r.t. manuscript in the track mode].

List of major amendments

 The title of the paper is slightly changed.

  1. Abstract has been modified.
  2. Introduction section is updated as per the Reviewers suggestions.
  3. Figures 1,2,3, and 6 have been updated.
  4. Seven new Reference have been added in the text.   
  5. Replies to comments of Reviewer #1:

    GENERAL COMMENTS: 

    Brief summary

    This paper, entitled ‘Assessing Vulnerability of Water Balance to Climate Change: Implications for Sustainable Water Future’ this study focuses on assessing the impact of climate change on the water balance components (precipitation, surface runoff, groundwater flow, percolation, etc.) at the river basin scale in a humid tropical region (the Periyar river basin, Kerala, India) using the SWAT hydrological and three general circulation models (GCMs: CanESM5, CNRM-CM6-1, and MPI-ESM1-2-LR ) under two shared socio-economic pathways (SSP 245 and SSP 585) emission scenarios.

    The authors concluded that the climate change would result in an increase in precipitation and reduction in evapotranspiration, on one hand, and a significant rise in runoff, streamflow, and percolation, on the other hand, through mid (2041–2070) and far (2071–2100) future periods. 

    Such situations often give rise to floods and affect the hydrology. They considered that alteration in the hydrological variables considerably affect the groundwater re charge rate, soil moisture, and crop water requirement, which may create water stress situations. Therefore, they recommended that the developed approaches could assist decision makers in developing strategies for the long-term management of water resources.

    The list of literature is very well balanced and updated.

    Ans: We thank the Reviewer for reviewing our work and providing the valuable comments and suggestions. The authors have tried their best to address all the comments and the response to the specific comments have been addressed below.

    COMMENT 1: Before line 97, you may add a paragraph on the related literature on the specific climate change impacts on the water resources in the study country/region ;

    Ans: Thanks for this suggestion. The changes have been incorporated in the manuscript.

          Refer Line no. 113–123.

          “Humid tropical regions have been a point of attraction for human settlements due to its unique ecohydrology [23], [24]. The Western Ghats, a humid tropical region in India, is one such region that has been experiencing various changes in water resources due to climate change. This includes changes in precipitation pattern with increasing trend in southern part of the region whereas decreasing trend in the upper part of this region [25]. This contrasting trend in Southwest monsoon rainfall in the northern and southern Western Ghats has been reported in other studies as well [26]. The weakening of the vertical velocity and reduced summer mean rainfall over the orographic region has also been reported [27]. Other than this, certain regional studies have reported this area to be vulnerable to water scarcity [28] whereas in some regions there has been an increase in flooding events [29].”

    COMMENT 2: It seems that you based your future projections using the LULC of 2016. Why you haven’t used the LULC projections you developed in your paper published in Water in 2022?

    Ans: We thank the Reviewer for this question. The present study uses CMIP6 General Circulation Models (GCMs) climate projections in which landuse Model Intercomparison Project (MIP) is included (Eyring et al. 2016) which was not proposed in earlier version of CMIP5. So, in order to avoid repetition of the landuse effect, only climate change impact based on SSP scenarios are considered in this study.

    Also, there are several studies that have shown that the impact of climate change is dominant over LULC change (Sinha et al. 2020 [48], Chanapathi et al. 2020 [49]) so in this study the main focus is over analysing the impact of climate change on water balance components based on latest CMIP6 SSP scenarios.

    COMMENT 3: For the discharge data at the outlet: i) Have you taken into consideration the storage and release from the dams?; ii) you have data from 2000 to 2015 and in the your above mentioned your paper you have also data from 1991, why did you use for the calibration/validation only the periods : 2000-2004 and 2006-2010 ?

    Ans: We thank the Reviewer for the comments. For the discharge data at outlet, the storage from Idamalayar reservoir has been accounted. However, for Idukki reservoir, there is no water release into the Periyar river, instead, the water is directed outside the watershed into another river called Thodupuzha river (source: http:// www. kseb. in). Thus, the same has been implemented accordingly in the study and updated in the text. The details have been added in the study area description section.

          The reviewer is correct that the gauge discharge data for this station is available from 1991 onwards as mentioned in (Sadhwani et al. 2022) but this data is not continuous. So, the period from 2000–2004 and 2006–2010 are selected for calibration and validation respectively based on the availability of continuous gauge discharge data. The year 2005 is also neglected in this process due to inconsistencies observed in the input precipitation data which has also been reported in a study in the nearby region by Visweshwaran et al. (2022) [9].

          Refer Line no.: 166–170.

    The PRB is a complex watershed experiencing various constraints related to water distribution and reservoir operations. For instance, a major portion of the water stored in the Idukki reservoir is channelled outside the watershed. Further, a certain portion of the area comes under forest reserves which makes the area limited to several anthropogenic constraints. The details are discussed by Sadhwani et al. (2023) [33].”

    COMMENT 4: How you can explain the decrease of projected ET while both temperature and rainfall are increasing? 

    Ans: We thank the Reviewer for highlighting this point. It is rightly noticed by the reviewer that overall precipitation is increasing, still evapotranspiration (ET) is showing a decrease in magnitude. To understand this, it should be noticed that in the maximum increase in precipitation is observed in the sub-basins on the eastward side of the study area in all the future time segments (Figure 8-a). However, this region is having high slopes and thus a major part of the precipitation converts to runoff (Figure 9-a) and not much water is available for ET. Thus, the increase in ET is not very high in this region. The same trend has been observed in future time segments. Further, a major part of ET is contributed from the middle part of the watershed, where a decrease in precipitation is observed which results to lower ET values (Figure 8-a). For the westward part of the watershed, although there is an increase in precipitation but due to presence of high built-up area, the runoff is also relatively high, and less water is available for ET.

    Thus, an overall reduction in ET is observed over the basin.

    COMMENT 5: I think speaking of ‘water stress situations’ (line 440) with current and future abundant rain and runoff is not suitable particularly in this humid tropical region. Nevertheless, if you consider that there would be some  specific periods of water stress, I highly encourage you to add a section on scenario analysis : for example adding more dams, digging boreholes, etc.

    Ans: We thank the Reviewer for highlighting this part. The authors here meant to highlight water stress situation in specific sub-basins in the central part of the study area where both precipitation and runoff have shown decrease in near and mid future. However, we understand that the statement lacks clarity, so it has been updated accordingly.

          Refer Line no.   520–523.

          The findings of this study highlight the need for proactive and sustainable management of water resources, including irrigation requirements and groundwater discharge, to mitigate the negative effects of climate change and prevent water stress/surplus situations in the specific sub-basins.

    Minor comments

    COMMENT 6: No need of decimals as the study area is large and you are dealing of projections.

    Ans: As per the suggestion of the Reviewer the decimals from catchment area has been removed.

    Refer Line no. 140–142.

    “The Periyar river is the second-longest river (> 244 km in length) in Kerala, with a catchment area of approximately 4793 km2.”

    COMMENT 7: In fig.1: a) add : Arabian Sea, Western … Mountaind ; b) what is the difference between Plantation and Cropland ?; c) : inverse the colors (brown for high and blue for low)

    Ans: We thank the Reviewer for this suggestion. Figure 1-a has been updated with adding details regarding Arabian Sea. However, Western Ghats could not be added in the Figure due to space constrain and clear visibility.

          Here “plantation” means large area of tall trees of eucalyptus, teakwood, rubber and coconut planted over large farms that to some extent act as forests. Whereas, “cropland” means small types of crops of rice, millets, banana, coffee, and cardamon.

          In Figure 1-c, a new color combination (High: purple and Low: brown) has been applied for better visualization of elevation details.

          Refer Line no. 171–172.

    References:

    Chanapathi, Tirupathi, and Shashidhar Thatikonda. "Investigating the impact of climate and land-use land cover changes on hydrological predictions over the Krishna river basin under present and future scenarios." Science of the Total Environment 721 (2020): 137736.

    1. Eyring et al., “Overview of the Coupled Model Intercomparison Project Phase 6 (CMIP6) experimental design and organization,” Geosci. Model Dev., vol. 9, no. 5, pp. 1937–1958, May 2016.
    2. Sadhwani, T. I. Eldho, M. K. Jha, and S. Karmakar, “Effects of Dynamic Land Use/Land Cover Change on Flow and Sediment Yield in a Monsoon-Dominated Tropical Watershed,” Water, vol. 14, no. 22, p. 3666, Nov. 2022, doi: 10.3390/w14223666. [Online]. Available: http://dx.doi.org/10.3390/w14223666
    3. Visweshwaran, R. Ramsankaran, T. I. Eldho, and M. K. Jha, “Hydrological Impact Assessment of Future Climate Change on a Complex River Basin of Western Ghats, India,” Water, vol. 14, no. 21, p. 3571, Nov. 2022.

    Sinha, Rakesh Kumar, T. I. Eldho, and Ghosh Subimal. "Assessing the impacts of land use/land cover and climate change on surface runoff of a humid tropical river basin in Western Ghats, India." International Journal of River Basin Management (2020): 1-12.

    In text reference:

    1. E. Wohl et al., “The hydrology of the humid tropics,” Nat. Clim. Chang., vol. 2, no. 9, pp. 655–662, 2012.
    2. P. Hamel et al., “Watershed services in the humid tropics: Opportunities from recent advances in ecohydrology,” Ecohydrology, vol. 11, no. 3, p. e1921, Apr. 2018.
    3. CWC. Kerala Floods of August 2018. Central Water Commission, New Delhi. 2018. Available online: https://sdma.kerala.gov.in/ wp-content/uploads/2020/10/Kerala_28122018_CWC_December-2018.pdf.
    4. K. P. Sudheer et al., “Role of dams on the floods of August 2018 in Periyar River Basin, Kerala,” Curr. Sci., vol. 116, no. 5, pp. 780–794, 2019.
    5. Mohanakrishnan, A.; Verma, C.V.J. History ofthe Periyar Dam with Century Long Performance; Central Board of Irrigation & Power: New Delhi, India, 1997.
    6. D. Lu and Q. Weng, “A survey of image classification methods and techniques for improving classification performance,” Int. J. Remote Sens., vol. 28, no. 5, pp. 823–870, Mar. 2007.
    7. D. . Pai, M. Rajeevan, O. . Sreejith, B. Mukhopadhyay, and N. . Satbha, “Development of a new high spatial resolution (0.25° × 0.25°) long period (1901-2010) daily gridded rainfall data set over India and its comparison with existing data sets over the region,” MAUSAM, vol. 65, no. 1, pp. 1–18, Jan. 2014.

Reviewer 2 Report

This paper assesses the impact of climate change on various hydrological parameters (precipitation, surface 13 runoff, groundwater flow, percolation, streamflow) through the integration of General Circulation Models with the SWAR model using two shared socio-economic pathways, the SSP2 4.5 and SSP5 8.5.

These are my main comments:

1.       In Table 1, you mention the resolution of temperature is 1 x 1 degrees, but in the text it is 0.5.

2.       Fix subsection numbering in Methodology

3.       Line 196: you need to add a sentence or two about the bias correction technique used

4.       Figure 3: It is not clear what Ensemble means in the figures nor the methodology mentions how this is performed.

5.       Figures 4&5: Fix the legend

6.       You mention that precipitation is the primary source of runoff, streamflow, groundwater discharge. However, you don’t discuss or explain the outcome of SSP5 8.5 in Table 2 for the near future where precipitation is decreasing while there is an increase in streamflow, groundwater discharge.

7.       Line 377: how does the excess runoff leads to decreasing streamflow? Is this correct?

8.       Lines 397 and 398: However, regional characteristics play a major role in defining the behavior of water balance components. It will be good here to discuss specific characteristics of your region that differentiates it from other studies mentioned in the same paragraph, to know why the outcome is similar to some and not to other studies.

9.       One main thing your discussion and conclusion is missing is what makes your study significantly different from other studies done for the same climates and within the same region.

Author Response

Assessing Vulnerability of Water Balance to Climate Change in Humid Tropics: Implications for Sustainable Water Future

Kashish Sadhwani and T. I. Eldho

Article number: sustainability-2322136

The authors would like to appreciate the time and effort taken in the review of the manuscript by the Editor and reviewers. We are grateful for the valuable comments and suggestions, which have helped to improve the quality of the manuscript. Further, we thank all the anonymous reviewers for their positive comments. In the revised version, we have attempted to address all the concerns of the Editor and the reviewers.

Please note that the review comments are given in red font while the responses to these comments are given in blue font with “manuscript changes highlighted in Italics within quotes”.

[Line numbers mentioned in the responses are w.r.t. manuscript in the track mode].

List of major amendments

  1. The title of the paper is slightly changed.
  2. Abstract has been modified.
  3. Introduction section is updated as per the Reviewers suggestions.
  4. Figures 1,2,3, and 6 have been updated.
  5. Seven new Reference have been added in the text.

Replies to comments of Reviewer #2:

GENERAL COMMENTS:

This paper assesses the impact of climate change on various hydrological parameters (precipitation, surface runoff, groundwater flow, percolation, streamflow) through the integration of General Circulation Models with the SWAR model using two shared socio-economic pathways, the SSP2 4.5 and SSP5 8.5.

COMMENT 1: In Table 1, you mention the resolution of temperature is 1 x 1 degrees, but in the text it is 0.5.

Ans: We thank the Reviewer for identifying this point. The correct resolution for temperature data is 1 x 1 degree as mentioned in Table 1. Please consider it as a typing error and excuse this mistake. The correction has been made in the manuscript.

Refer Line no. 189.

COMMENT 2: Fix subsection numbering in Methodology

Ans: We thank the Reviewer for highlighting this error in the manuscript. All the headings have been rechecked properly and corrections has been made.

Refer Line no. 140, 174, 201, 217, 240, and 290.

COMMENT 3: Line 196: you need to add a sentence or two about the bias correction technique used

Ans: We thank the Reviewer for the suggestion. The details of the statistical downscaling method and bias correction method has been added in the text.

Refer Line No. 241–245.

“The future meteorological data inputs, including precipitation, and minimum and maximum temperature, were obtained from GCMs after statistical downscaling using a non-parametric kernel regression model [46] and then bias corrected with respect to historical IMD data using quantile delta mapping technique [42]. The details of the process applied is discussed in detail by Salvi et al. (2013) [42].”

COMMENT 4: Figure 3: It is not clear what Ensemble means in the figures nor the methodology mentions how this is performed.

Ans: We thank the Reviewer for highlighting this point. The term “ensemble” in Figure 3 refers to the ensemble average of the three GCMs used in the study. The details are updated in the text.

Refer Line no. 249–250.

“Finally, an equal weighted ensemble average of the GCM results is enumerated for all the three time segments.”

COMMENT 5: Figures 4&5: Fix the legend.

Ans: The legend has been updated.

Refer Line no. 273.

COMMENT 6: You mention that precipitation is the primary source of runoff, streamflow, groundwater discharge. However, you don’t discuss or explain the outcome of SSP5 8.5 in Table 2 for the near future where precipitation is decreasing while there is an increase in streamflow, groundwater discharge.

Ans: We thank the Reviewer for raising this mistake. Please consider it as typing error where the change in streamflow at outlet (Neeleshwaram gauging station) in SSP 585 scenario is mistakenly written as “+5.44%” instead of “-5.44%”. The corresponding precipitation, runoff, and groundwater flow also show a decrement in this sub-basin with decrease of -22%, -38%, and -37%, respectively. Thus, the streamflow at this site is decreasing in response to the decrease observed in precipitation, runoff, and groundwater flow to maintain the water balance.

It should be noted that the values of groundwater discharge into the stream noted in Table 2 is an average value of the entire basin.

Regarding the positive value of groundwater discharge into the stream. It has been checked properly and it is correctly mentioned. Since some sub-basins are showing high increase in groundwater discharge (eastern part of the study area), thus an overall an increase is observed in average groundwater discharge.

The discussion on the results of these segments has been added in the text.

Refer Line no.  409–419.

Both percolation and streamflow are expected to increase by +78.64% and +118.73%, respectively under SSP 245 whereas +76.52% and +113.84% under SSP 585 scenario, by 2100. Such changes are alarming and need to be understood better to manage water resources efficiently. It should be noted that here the outlet is considered at the location of Neeleshwaram gauging station (refer to Figure 1-a) to analyze the change in streamflow. This site was selected as the calibration for the model was done with respect to this location. Additionally, it should be noted that the change in streamflow is not determined by taking the average over the entire basin, as some sub-basins exhibit a positive change while others demonstrate a negative change, which could potentially nullify the overall change in streamflow.

COMMENT 7: Line 377: how does the excess runoff leads to decreasing streamflow? Is this correct?

Ans: Thank you for your question. We carefully reviewed the paper, but we could not find any mention of excess runoff leading to a decrease in streamflow. However, we would like to clarify a typing error in Table 2 and Line No 474 where change in streamflow at outlet (Neeleshwaram gauging station) in SSP 585 scenario is mistakenly written as “+5.44%” instead of “-5.44%”. This mistake has been corrected.

It shall be noticed that in this case, the precipitation has decreased in this sub-basin which resulted in reduced runoff and thus a reduction in streamflow has been observed for this specific sub-basin.

Refer Line No. 420 and Figure 9–a.

COMMENT 8: Lines 397 and 398: However, regional characteristics play a major role in defining the behavior of water balance components. It will be good here to discuss specific characteristics of your region that differentiates it from other studies mentioned in the same paragraph, to know why the outcome is similar to some and not to other studies.

Ans: Thank you for this suggestion. A discussion on the watershed characteristics and its effect on the water balance components has been added in the text.

Refer Line no. 429–447.

The spatial pattern of changes in the water balance drivers showed a strong correlation among the components. Other than this, the topographical factor also plays an important role in this study area. The sub-basins on the eastern side of the watershed has been experiencing high runoff (Figure 9–a) in response to the combined effect of high slopes and increased precipitation (Figure 1–c and 8–a). Whereas the sub-basins in the central part of the with relatively lesser slopes experience reduced runoff due to direct effect of reduced precipitation (Figure 1–c and 8–a). Contrary to this, the sub-basins in the western end near the coast, are expected to experience high runoff with little increase in precipitation in future (Figure 9–a and 8–a) despite having a flat terrain. This behaviour can be related to the landuse characteristics of this area, as it is highly covered with built-up area (Figure 1–b) and even a small amount of precipitation leads to high runoff values in such regions. Thus, landuse characteristics play a major role in watershed management. Since excess runoff often leads to increased river flow and result as a potential cause of flooding. Thus, it is crucial to give special attention to this aspect in water resource management and flood control. Overall, with precipitation being identified as the main driver component for the other hydrological changes, it is interesting to note that the changes in hydrology can be anticipated early. Thus, preparedness against climate change action can be done efficiently and effectively.

COMMENT 9: One main thing your discussion and conclusion is missing is what makes your study significantly different from other studies done for the same climates and within the same region.

Ans: We thank the Reviewer for this question. The main significance of this study is the uniqueness of the Periyar river basin which has a long river (length> 244 km) with three major reservoirs along its course subjected to several operational constraints. The Western Ghats has several rivers, but a very few studies have been conducted on river basins with active reservoirs. Other than this, the unique climatology of this region in combination with the topographical characteristics adds insights to the combined response of this combination to climate change. Only a few studies are available on this topic dealing with such big watersheds subjected to such anthropogenic constraints. Further, in this study, the impact of climate change on water balance components has been tested with the latest CMIP6 based SSP climate change scenarios which has not been applied on this region with such combination in previously available studies.

The details regarding the features of the study area and a comparison with other studies has been added in the text.

Refer Line no. 166–170, 483–490 and 530–533.

“The PRB is a complex watershed experiencing various constraints related to water distribution and reservoir operations. For instance, a major portion of the water stored in the Idukki reservoir is channelled outside the watershed. Further, a certain portion of the area comes under forest reserves which makes the area limited to several anthropogenic constraints. The details are discussed by Sadhwani et al. (2023) [33].”

“Further, the outcomes of this study were also compared with the results of Sadhwani et al. (2023) [33] where the earlier climate models with CMIP5 projection were tested in the Periyar river basin. It was identified that the results were in consensus with the earlier study with runoff showing a similar spatial change across the basin with little change in magnitude. Thus, this study also adds insights about the similarity between the outcomes of CMIP5 and CMIP6 climate models in this study area.”

Further, this study can also be considered as a reference for long river catchments with varied topographical changes and reservoir action constraints in order to deal with climate change issues.

References:

[33] Sadhwani K, Eldho TI, Karmakar S. “Investigating the influence of future landuse and climate change on hydrological regime of a humid tropical river basin”. Environmental Earth Sciences. May 2023; 82(9):210.

Reviewer 3 Report

Dear authors,

please find comments in the attachment.

Author Response

Assessing Vulnerability of Water Balance to Climate Change in Humid Tropics: Implications for Sustainable Water Future

Kashish Sadhwani and T. I. Eldho

Article number: sustainability-2322136

The authors would like to appreciate the time and effort taken in the review of the manuscript by the Editor and reviewers. We are grateful for the valuable comments and suggestions, which have helped to improve the quality of the manuscript. Further, we thank all the anonymous reviewers for their positive comments. In the revised version, we have attempted to address all the concerns of the Editor and the reviewers.

Please note that the review comments are given in red font while the responses to these comments are given in blue font with “manuscript changes highlighted in Italics within quotes”.

[Line numbers mentioned in the responses are w.r.t. manuscript in the track mode].

List of major amendments

  1. The title of the paper is slightly changed.
  2. Abstract has been modified.
  3. Introduction section is updated as per the Reviewers suggestions.
  4. Figures 1,2,3, and 6 have been updated.
  5. Seven new References have been added in the text.

Replies to comments of Reviewer #3:

GENERAL COMMENT:  

The manuscript provides an interesting understanding into the future impact of the water balance availability in the humid climate conditions adopting a multi-model ensemble for forecast optimalization efficacy. However, the size of the river basin is too big, and the study outputs are too coarse. Is there inter - annual variability in the water balance course.

Ans: We thank the reviewer for time and effort given in reviewing the manuscript and providing valuable comments to improve the quality of the content. We acknowledge your concern regarding the size of the river basin and the coarse resolution of the study outputs. We understand that the size of the river basin examined in our study may not provide a detailed analysis of specific localized conditions at very fine resolution. However, the chosen scale allows us to capture the broader patterns and trends in the water balance dynamics within the region of interest. Also, the spatial change across the study area as discussed in Figure 7–9 adds information regarding the impact of climate change in specific sub-basins which will be helpful in designing management strategies for the specific sub-basins. Also, since the management policies are applied at Taluk level (minimum administrative unit) the available resolution of sub-basins will be suitable to implement the management changes.

The reviewer is right regarding the inter-annual variability in water balance course. The inter-annual changes for precipitation, ET, runoff, and groundwater discharge were observed for the period 1989–2019 (refer Figure below). The results show maximum variability across the months of June to September during monsoon season.

Figure: Inter-annual variability in monthly precipitation (mm), ET (mm), runoff (mm), and groundwater discharge (mm) in PRB for the period 1989–2019.

COMMENT 1: Which period is affected by the climate change the most?

Ans: The seasonal variability is not considered in this study. However, our past study by Sadhwani et al. (2023) [33] showed that have shown the impact of climate change will be maximum in the monsoon season (June to September).

COMMENT 2: During which months the changes in the water balance the most dramatic? The study should point out the most problematic season of the year.

Ans: The monsoon season months from June to September are the most problematic season as precipitation for the entire year is concentrated over these months. It is mentioned in discussion section.

Refer Line no. 489–490.

“Furthermore, it was also noticed that the maximum impact of climate change on water balance occurred during the monsoon season (June to September).

COMMENT 3: The title of the manuscript is too general. It should be clearly described that it is concerning to the water balance in humid climate conditions. The title of the study should be modified to make it clearer.

Ans: The title has been updated as per the Reviewer’s suggestion to provide more clarity regarding the context of the research.

Refer Line No. 2–3

“Assessing Vulnerability of Water Balance to Climate Change in Humid Tropics: Implications for Sustainable Water Future”

COMMENT 4: The abstract is too general. Could you, please, provide the more consistent abstract, please? Summarize all the information from the whole study results and key findings to attract readers.

Ans: Thank you for this suggestion. The abstract has been updated accordingly.

Refer Line 8–32.

“Sustainability in hydrology aims at maintaining a high likelihood of meeting future water demands without compromising hydrologic, environmental, or physical integrity. Many factors, such as the effects of climate change and the competing priorities of different stakeholders, introduce uncertainty into water resource systems on a large scale. Thus, understanding the impact of global climate change on hydrology and water balance at the local scale is vital. This study focuses on assessing the impact of climate change on the water balance components (precipitation, surface runoff, groundwater flow, percolation, etc.) at the river basin scale in a humid tropical region. The Periyar river basin (PRB) in Kerala in India is considered as a case study and the SWAT hydrological model was adopted to obtain the water balance components. Three general circulation models (GCMs: CanESM5, CNRM-CM6-1, and MPI-ESM1-2-LR ) are considered under two shared socioeconomic pathways (SSP 245 and SSP 585) emission scenarios to assess the impact of climate change till 2100. For PRB, the results demonstrate a significant increase in streamflow (>65%) and runoff (>40%) in the mid (2041–2070) and far (2071–2100) future under both the SSP scenarios, indicating a potential vulnerability to future floods. Conversely, in the near future under SSP 585, a decrease in runoff (-15%) and nominal changes in streamflow (-5%) are observed. Spatially, the eastern sub-basins and the west coast of the Periyar river basin are projected to experience higher precipitation events, while the central region faces reduced precipitation and low flow rates. The findings emphasize the need for proactive and sustainable management of water resources, considering irrigation requirements, groundwater discharge, and flood control measures, to mitigate the negative effects of climate change and prevent water stress/surplus situations in specific sub-basins. This study contributes to our understanding of the potential impacts of climate change on water balance and stresses the significance of sustainable water resources management to effectively respond to these challenges. By integrating scientific knowledge into policy and management decisions, we can strive towards a resilient water future in the context of a changing climate.”

COMMENT 5: Would you be so kind to describe, what is the novelty of this study? This information is missing within the text, and it must be included there.

Ans: We thank the Reviewer for this question. The main significance of this study is the uniqueness of the Periyar river basin which has a long river (length> 244 km) with three major reservoirs along its course subjected to several operational constraints. The Western Ghats, which is a biological hotspot identified by UNESCO, has several rivers, but a very few studies have been conducted on river basins with active reservoirs. Other than this, the unique climatology of this region in combination with the topographical characteristics adds insights to the combined response of this combination to climate change. Only a few studies are available on this topic dealing with such big watersheds subjected to such anthropogenic constraints. Further, in this study, the impact of climate change on water balance components has been tested with the latest CMIP6 based SSP climate change scenarios which has not been applied on this region with such combination in previously available studies.

The details regarding the features of the study area and a comparison with other studies has been added in the text.

Refer Line no. 166–170, 483–489 and 530–533.

“The PRB is a complex watershed experiencing various constraints related to water distribution and reservoir operations. For instance, a major portion of the water stored in the Idukki reservoir is channelled outside the watershed. Further, a certain portion of the area comes under forest reserves which makes the area limited to several anthropogenic constraints. The details are discussed by Sadhwani et al. (2023) [33].”

“Further, the outcomes of this study were also compared with the results of Sadhwani et al. (2023) [33] where the earlier climate models with CMIP5 projection were tested in the Periyar river basin. It was identified that the results were in consensus with the earlier study with runoff showing a similar spatial change across the basin with little change in magnitude. Thus, this study also adds insights about the similarity between the outcomes of CMIP5 and CMIP6 climate models in this study area.”

Further, this study can also be considered as a reference for long river catchments with varied topographical changes and reservoir operation constraints in order to deal with climate change issues.

COMMENT 6: The more detailed information should be incorporated concerning to the size and shape of the river basins, their exploitations, population density as well as the type of the climate…, because now it is too general and vague.

Ans: We thank the Reviewer for this suggestion. Additional details regarding the geography and population density of the study area has been updated in the manuscript.

Refer Line no. 142–145, 147–151, and 153–170.

The basin has an inverted 'L' shape, and its overall drainage pattern is dendritic in nature. It is a west flowing river that originates at an elevation of 2438 m above mean sea level (AMSL) in the Western Ghats Mountain range and drains into the Arabian Sea.”

“The PRB has a tropical-humid climate, with rainfall concentrated over the months of July through November. The mean annual rainfall in PRB is 3200 mm (CWC, 2018) [30], and the maximum and minimum temperature range from 25Ëš C to 32Ëš C and 14Ëš C to 19Ëš C, respectively in the basin [31].”

” The forest area mainly consists of tropical evergreen trees and plantation predominantly features coconut and areca nut. The agro-climatic conditions of the region are favourable for cultivation of cash crops including coffee, pepper, and cardamon which are the main source of income for the area. Periyar river, being a perennial river, is a vital source of water in the central parts of Kerala serving a population of more than 43,91,362 people (Census 2011, https://censusindia.gov.in/, accessed on 1 May 2023). There are three major reservoirs: Mullaperiyar Dam (capacity: 443.23 x106 m3), Idukki Dam (capacity: 5550 x106 m3), and Idamalayar Dam (capacity: 1089 x106 m3) (https://www.kseb.in/, (accessed on 15 February 2019), [32]) and one hydrological observation station at Neeleshwaram (10Ëš 12’ N 76Ëš 5’ E) in the basin (Figure 1-a). The average annual runoff from Neeleshwaram gauging station is recorded as 6686 x106 m3 [31]. The dams serve the purpose of electricity generation, flood control, and fulfilling irrigation water demands in the region. The PRB is a complex watershed experiencing various constraints related to water distribution and reservoir operations. For instance, a major portion of the water stored in the Idukki reservoir is channelled outside the watershed. Further, a certain portion of the area comes under forest reserves which makes the area limited to several anthropogenic constraints. The details are discussed by Sadhwani et al. (2023) [33].”

COMMENT 7: You are describing the water balance, but the most important information about the precipitation, temperature, evapotranspiration and discharge course are missing in the study area description part. Could you, please specify it?

Ans: We thank the Reviewer for this suggestion. The details regarding the average annual evapotranspiration and discharge have been added into the text.

Refer Line no. 147–152 and 163–164.

“The PRB has a tropical-humid climate, with rainfall concentrated over the months of June through November. The mean annual rainfall in PRB is 3200 mm (CWC, 2018) [30], and the maximum and minimum temperature range from 25Ëš C to 32Ëš C and 14Ëš C to 19Ëš C, respectively in the basin [31]. The average annual evapotranspiration of the basin is approximately 850 mm.”

“The average annual runoff from Neeleshwaram gauging station is recorded as 6686 x106 m3 [31].”

COMMENT 8: The plantation covers about 52.02% of the river basin, but what is it exactly, which crop? Could you specify the individual crops in detail?

Ans: The details regarding the plantation and cropping pattern in PRB has been added in the text.

Refer Line no. 153–156.

“The forest area mainly consists of tropical evergreen trees and plantation predominantly features rubber, eucalyptus, teakwood, coconut and arecanut. The agro-climatic conditions of the region are favourable for cultivation of cash crops including rice, millets, coffee, pepper, and cardamon which are the main source of income for the area.”

COMMENT 9: Could you add the model efficiency in the Figure 3, please? It would be more comprehensive for comparation of the results.

Ans: We thank the Reviewer for this suggestion. The Figure 3 has been updated with the model efficiency considering corelation coefficient (r) values.

Refer Line no. 267.

COMMENT 10: Could you use the column chart for precipitation in Figure 8? It would be more comprehensive.

Ans: We thank the Reviewer for the suggestion for the changes in Figure 8. As it is a spatial variation figure, it will be difficult to explain the spatial variation of precipitation across the watershed for all sub-basins using column charts. As distinct colors for sub-basins are given with the legend, we hope that the details can be easily understood.

COMMENT 11: Identify the main outcomes on the base of the study results. Which measures should be adopted to prevent the negative impacts resulting from this study? Are the positive effects of the climate change resulting from this study?

Ans: Thank you for these suggestions. The outcomes of the study show a spatial variation in hydrological characteristics across the basin in response to climate change. As some sub-basins were identified to experience excess precipitation leading to excess runoff which may lead to floods while other sub-basins showed less precipitation leading to reduced runoff. Thus, appropriate measures are required in respective areas, with regions showing reduction in water availability (reduced precipitation) must apply methods to deal with the water scarcity that may rise in future. Thus, there is positive as well as negative effect due to climate change and need to be dealt accordingly.

Refer Line no. 523–530.

This study also underscores the significance of using modeling approaches in developing long-term strategies for water resource management that can support sustainable development in the face of climate change. Some of these strategies may include flood control in areas where excess runoff is observed. While others may include utilizing this information on excess runoff water availability in future to enhance productivity by hydro-power generation, agricultural productivity, increase water recreation and tourism. Similarly for regions reflecting reduced water availability in the future must be dealt accordingly to utilize the water resources more effectively.

COMMENT 12: Could you provide the references from the recent studies?

Ans: New references have been added in the text.

Refer References section.

References:

[25]      N. Chandu, T. I. Eldho, and A. Mondal, “Hydrological impacts of climate and land-use change in Western Ghats, India,” Reg. Environ. Chang., vol. 22, no. 1, p. 32, Mar. 2022.

[26]      H. Varikoden, J. V. Revadekar, J. Kuttippurath, and C. A. Babu, “Contrasting trends in southwest monsoon rainfall over the Western Ghats region of India,” Clim. Dyn., vol. 52, no. 7–8, pp. 4557–4566, Apr. 2019.

[27]      K. Rajendran, A. Kitoh, J. Srinivasan, R. Mizuta, and R. Krishnan, “Monsoon circulation interaction with Western Ghats orography under changing climate,” Theor. Appl. Climatol., vol. 110, no. 4, pp. 555–571, Dec. 2012.

[28]      T. M. Sharannya, K. Venkatesh, A. Mudbhatkal, M. Dineshkumar, and A. Mahesha, “Effects of land use and climate change on water scarcity in rivers of the Western Ghats of India,” Environ. Monit. Assess., vol. 193, no. 12, p. 820, Dec. 2021.

[29]      KSCSTE, Statement of Climate For the State of Kerala: 2021. 2021.

[33]      K. Sadhwani, T. I. Eldho, and S. Karmakar, “Investigating the influence of future landuse and climate change on hydrological regime of a humid tropical river basin,” Environ. Earth Sci., vol. 82, no. 9, p. 210, May 2023.

[46]      S. Kannan and S. Ghosh, “A nonparametric kernel regression model for downscaling multisite daily precipitation in the Mahanadi basin,” Water Resour. Res., vol. 49, no. 3, pp. 1360–1385, Mar. 2013

Round 2

Author Response

Assessing the Vulnerability of Water Balance to Climate Change at a River Basin Scale in the Humid Tropics: Implications for a Sustainable Water Future

Kashish Sadhwani and T. I. Eldho

Article number: sustainability-2322136

The authors would like to express their gratitude for the dedicated time and effort invested by the Editor and reviewers in reviewing the manuscript. We specially thank the two reviewers We sincerely thank the two reviewers for accepting the manuscript. Also, we appreciate the valuable comments and suggestions provided, which have significantly contributed to enhancing the overall quality of the manuscript. Additionally, we extend our thanks to all the anonymous reviewers for their positive feedback and constructive input.

In this revised version, we have attempted to address all the concerns of the Editor and the reviewer. The new changes in second revision are highlighted with yellow color in the manuscript to separate it from previous corrections.

Please note that the review comments are given in red font while the responses to these comments are given in blue font with “manuscript changes highlighted in Italics within quotes”.

[Line numbers mentioned in the responses are w.r.t. manuscript in the track mode].

List of major amendments

  1. The title of the paper is slightly changed.
  2. Abstract has been modified.
  3. Introduction section is updated as per the Reviewers suggestions.
  4. A supplementary file has been added.

Replies to comments of Reviewer #3:

COMMENT 1: Summary:

Although the text of the article has been corrected and supplemented, the essential parts of the article remain unclear. In the manuscript is missing information about the river basins and its three major reservoirs: Mullaperiyar Dam (capacity: 443.23 x106 m3), Idukki Dam (capacity: 5550 x106 m3), and Idamalayar Dam (capacity: 1089 x106 m3). How is the increased evaporation from the water level of these damps considered in the study? How will the changes in the water balance affect the crop production and land use in the study area? This must be incorporated and included in the study.

Ans: We thank the Reviewer for these comments.

a). Additional information regarding the reservoir has been added in the text.

Refer Line No. 174–176.

“The Mullaperiyar Dam serves the purpose to divert water to the eastward side in the rain shadow region whereas the Idamalayar and Idukki dams major purpose is to generate electricity for the region.”

b). To consider the evaporation at the surface water of the reservoirs and lakes, SWAT has an inbuilt function that calculates the evapotranspiration based on the equation:

where  is the volume of water removed from the water body by evaporation during the day,  is an evaporation coefficient (0.6), E0 is the potential evapotranspiration for a given day, and SA is the surface area of the water body (ha) (Neitsch, Arnold, Kiniry, & Williams, 2011).

All these information was added in the model during SWAT model setup and its effect has been considered in the entire modelling process.

c). The effect of changes in water balance on crop production has not been considered in this study. However, changes in water balance components can affect crop production and land use in many ways. The impacts of climate change on water balance will lead to alterations in various aspects such as soil water storage, groundwater levels, and soil moisture status. These changes can significantly influence the irrigation requirements and quantity. Further, variations in precipitation and evapotranspiration, consequently, results in fluctuations in soil moisture status. Thus, these changes influence the cropping pattern or leading to shifting to other locations which subsequently affect the landuse pattern.

A discussion on the impact of changes in water balance components in PRB on crop yield has been added in “Discussion section”.

Refer Line No 472–486.

“An additional aspect of climate change impact on water balance components is the influence on crop productivity and yield. In the PRB, most of the agricultural area is limited near the downstream end (Figure 1–b). Projections indicate that in these sub-basins there is a notable decrease in precipitation in near future whereas a nominal decrease in mid and far future (Figure 8–a). The same trend is observed for ET, runoff, and groundwater discharge (Figure 8–b, 9–a, and 9–b). These changing trends raise concerns as reduced precipitation not only diminishes crop yield but also hampers crop growth. Additionally, runoff plays a crucial role in carrying nutrients from the soil and maintaining soil moisture. Consequently, a decrease in runoff can result in reduced nutrient availability for plant uptake and a decline in soil moisture content. The reduced flow of groundwater is also worrisome, as it can increase the risk of soil salinity and negatively impact soil fertility. However, it is essential to recognize that these changes are multifaceted, requiring further investigation into the associated agricultural risks within the region. Consequently, strategic planning and effective water resource management are imperative to address these challenges and mitigate their potential impacts on agriculture.”

COMMENT 2: Review:

It is a very difficult to discover the main findings and outputs of the study because the river basin is too big and the scale too course. It would a very useful to summarize all the results in the table, where the individual sub-basins will be compared, included their percentual changes of the water balance. Otherwise, the main findings are extinguished. Therefore, I recommend accepting the manuscript after minor revision.

Ans: We thank the Reviewer for this comment.

The results for all the sub-basins has been added in supplementary data and the information is mentioned in the text.

Refer Line No 419–421 and Supplementary Material.

Further, the results for all the above-mentioned water balance components, along with percolation and streamflow, averaged over the basin, are mentioned in Table 2 and the results for each sub-basins is mentioned in the Supplementary material (Table S1–S4).

COMMENT 3: Title

Although the title of the article has been slightly modified, it is still too general. In the title should be clearly specify which water balance is considering. Is it a water balance of a system, soil profile or a river basin? I recommend specifying in the title global climate change instead of the climate change. In addition, the term: “sustainable water future” is a very general: could you describe if it concerns the quality of quantity?

Ans: We thank the Reviewer for this suggestion. The title has been changed mentioning that the water balance is considered at river basin scale. Further, the study considers climate change impact at regional scale therefore we suppose that adding the term “Global” climate change may be misleading for readers.

In addition to this, the quantified assessment of climate change's impact on water balance components is a crucial aspect to be considered when planning and managing for a sustainable future. This objective has been explicitly stated in the study's aim within the introduction section, which helps the readers to understand the gist of the research content and also avoids a lengthy title.

Refer Line No. 2–4

“Assessing the Vulnerability of Water Balance to Climate Change at River Basin Scale in Humid Tropics: Implications for a Sustainable Water Future”

COMMENT 4: Abstract: The abstract exceeds the maximum amount of the words, see the journal sustainability requirements: the abstract should be a total of about 200 words maximum.

The text of the manuscript contains typos, e.g.:

Line 105: „is” instead of “ils”

Ans: Thank you for this comment.

The abstract size has been reduced and all other corrections has been made.

Refer Line no. 9–33 and 107.

COMMENT 5: Introduction

Would you be so kind to describe, what is the novelty of this study? This information is still missing within the text, and it must be included there.

Ans: Thank you for this comment.

This study discusses the impact of climate change on the water balance components at river basin scale in a humid tropical region based on the latest SSP climate change scenarios.  The Periyar river basin is considered in this study which is a complex watershed impaired with various anthropogenic constraints including controlled reservoir operation and restrictions on landuse changes. In addition, the unique climatic conditions due to orography influence makes it an important area for study. No past study has been done on this area to consider climate change impact on water balance with the latest SSP climate change scenarios. The details have been added in the text.

Refer Line no 135–141.

Thus, assessing the impact of future climate change, considering latest SSP scenarios, on water balance components within this complex watershed is the main focus of this study. This aspect has not been previously addressed in any other study in this region. The SWAT model is employed to conduct this assessment, allowing for a comprehensive evaluation of the effects. The datasets from three GCMs are utilized to analyze the change till 2100 under the SSP 245 and SSP 585 scenarios.

COMMENT 6: Materials and Methods/ Study area description and input data details/ Study area description

The study is describing the water balance of the river basin. However, the main and important information such as the hydrological water regime of the river basin and its descriptions is missing yet. That is crucial and necessary. When is the highest discharges in which period? What about the average annual temperature?

Ans: We thank the Reviewer for this suggestion.

The components of hydrological water regime including annual precipitation, evapotranspiration, surface runoff, and groundwater discharge for PRB are discussed in Section 3.3. (Refer Line No. 325–338)

A brief discussion on the water balance components is mentioned in the study area description. (Refer Line No. 153–157 and 169–172)

Since it is a large-elongated basin, the average annual temperature varies from 28Ëš C (near upstream) to 30Ëš C (near downstream end). This information has been added in the text.

Additional information on maximum daily flow is also added in the text.

Refer Line No. 155–156 and 170–172.

Since it is a large-elongated basin, the average annual temperature varies from 28Ëš C near upstream to 30Ëš C near the downstream end.”

“The maximum daily discharge measured between 1989 and 2017 peaked at 6324 m3/s. Notably, the last week of July or the first and second weeks of August experience the majority, or around 80% of the daily discharges over 2000 m3/s.”

COMMENT 7: Topographical and meteorological data

Could you describe why the specific data sets have been chosen? Do you have some special reason for that?

Ans: Thank you for this comment.

These datasets were chosen based on the requirement of the SWAT model and further on the basis of the availability and suitability of the data in the study area in reference to past literature. The same has been mentioned in the text.

Refer Line No. 230–233 and 216–217.

“SWAT input requires physical parameters, including soil type, LULC, DEM, and meteorological variables, including minimum and maximum temperature, solar radiation, precipitation, and wind speed.”

“These GCMs were selected on the basis of their ability to replicate the Indian Summer Monsoon [40] and performance in past studies [41].”

COMMENT 8: GCM climate data

Could you describe which scenario is the most favourable and what are the differences between the models?

Ans: We thank the Reviewer for this comment. The SSP 245 corresponds to the development pathways consistent with historical patterns and with medium radiating forcings (up to 4.5 W/m2) and SSP 585 scenario corresponds to high radiating forcings (up to 8.5 W/m2) till 2100. It has been mentioned in the text.

It is difficult to comment on the most favorable scenario as consequences of climate change vary with region and the effects are highly uncertain. However, for general comment, the SSP 585 proposes the most unfavorable climatic changes with high temperature rise and subsequent effects.

Refer Line No 219–227.

“In this study, two socioeconomic pathways: SSP 245 and SSP 585 were considered for analysis from each GCM. The SSP 245 corresponds to the development pathways consistent with historical patterns and with medium radiating forcings (up to 4.5 W/m2) in 2100. This scenario represents the medium range of the possible future. Whereas the SSP 585 corresponds to development pathways representing high industrial/economic growth and fossil fuel resources consumption with minimal efforts over reducing environmental concerns. This scenario corresponds to high radiating forcings (up to 8.5 W/m2) in 2100 [18] with potentially high challenges to mitigation strategies.

COMMENT 9: Results

Would you be so kind to summarize all the main findings in the more comprehensive way, please? Especially part “3.3. Impact of climate change on water balance components” is very unclear. Could you, please summarize all the parts of the water balance, please?

Ans: Thank you for this comment.

In Section 3.3, the results for the impact of climate change on the water balance components across the sub-basins have been discussed in a detail. The overall change averaged over the PRB has been tabulated in Table 2.

The discussion on the observed changes has been done in Section 3.4.

The addition of Supplementary data representing the change across all the sub-basins will help the readers to better understand the results in a comprehensive way.

These finding highlights the main result of the research, demonstrating the spatial variability of responses to climate change across the sub-basins in large river basin, despite their similar climatic conditions. This has been added in the conclusions.

Refer Line no. 419–421 and 544–546.

This finding highlights the main result of the research, demonstrating the spatial variability of responses to climate change across the sub-basins in a large river basin, despite their similar climatic conditions.

COMMENT 10: Concussions

How the results will be used? What was the main purpose of the study?

Ans: Thank you for this comment.

The main purpose of this study is to highlight the insights about the possible changes in the water balance components due to climate change at a river basin scale. This will help in identifying water scarce/ surplus regions and having this information would supplement to decision making in management processes. Furthermore, having the knowledge of these probable changes, specific strategies can be planned by government agencies, NGO’s, agricultural industry, and other stakeholders.

This would encourage for optimum water consumption practices, allowing for better management and allocation of water resources.

Refer Line No. 553–558.

This study also underscores the significance of using modeling approaches in developing long-term strategies for water resource management that can support sustainable development in the face of climate change. Some of these strategies may include flood control in areas where excess runoff is observed. While others may include utilizing this information on excess runoff water availability in future to enhance productivity by hydro-power generation, agricultural productivity, increase water recreation and tourism.

References

Neitsch, S. L., Arnold, J. G., Kiniry, J. R., & Williams, J. R. (2011). Soil and water assessment tool theoretical documentation version 2009. Texas Water Resources Institute.

Assessing the Vulnerability of Water Balance to Climate Change at a River Basin Scale in the Humid Tropics: Implications for a Sustainable Water Future

Kashish Sadhwani and T. I. Eldho

Article number: sustainability-2322136

The authors would like to express their gratitude for the dedicated time and effort invested by the Editor and reviewers in reviewing the manuscript. We specially thank the two reviewers We sincerely thank the two reviewers for accepting the manuscript. Also, we appreciate the valuable comments and suggestions provided, which have significantly contributed to enhancing the overall quality of the manuscript. Additionally, we extend our thanks to all the anonymous reviewers for their positive feedback and constructive input.

In this revised version, we have attempted to address all the concerns of the Editor and the reviewer. The new changes in second revision are highlighted with yellow color in the manuscript to separate it from previous corrections.

Please note that the review comments are given in red font while the responses to these comments are given in blue font with “manuscript changes highlighted in Italics within quotes”.

[Line numbers mentioned in the responses are w.r.t. manuscript in the track mode].

List of major amendments

  1. The title of the paper is slightly changed.
  2. Abstract has been modified.
  3. Introduction section is updated as per the Reviewers suggestions.
  4. A supplementary file has been added.

Replies to comments of Reviewer #3:

COMMENT 1: Summary:

Although the text of the article has been corrected and supplemented, the essential parts of the article remain unclear. In the manuscript is missing information about the river basins and its three major reservoirs: Mullaperiyar Dam (capacity: 443.23 x106 m3), Idukki Dam (capacity: 5550 x106 m3), and Idamalayar Dam (capacity: 1089 x106 m3). How is the increased evaporation from the water level of these damps considered in the study? How will the changes in the water balance affect the crop production and land use in the study area? This must be incorporated and included in the study.

Ans: We thank the Reviewer for these comments.

a). Additional information regarding the reservoir has been added in the text.

Refer Line No. 174–176.

“The Mullaperiyar Dam serves the purpose to divert water to the eastward side in the rain shadow region whereas the Idamalayar and Idukki dams major purpose is to generate electricity for the region.”

b). To consider the evaporation at the surface water of the reservoirs and lakes, SWAT has an inbuilt function that calculates the evapotranspiration based on the equation:

where  is the volume of water removed from the water body by evaporation during the day,  is an evaporation coefficient (0.6), E0 is the potential evapotranspiration for a given day, and SA is the surface area of the water body (ha) (Neitsch, Arnold, Kiniry, & Williams, 2011).

All these information was added in the model during SWAT model setup and its effect has been considered in the entire modelling process.

c). The effect of changes in water balance on crop production has not been considered in this study. However, changes in water balance components can affect crop production and land use in many ways. The impacts of climate change on water balance will lead to alterations in various aspects such as soil water storage, groundwater levels, and soil moisture status. These changes can significantly influence the irrigation requirements and quantity. Further, variations in precipitation and evapotranspiration, consequently, results in fluctuations in soil moisture status. Thus, these changes influence the cropping pattern or leading to shifting to other locations which subsequently affect the landuse pattern.

A discussion on the impact of changes in water balance components in PRB on crop yield has been added in “Discussion section”.

Refer Line No 472–486.

“An additional aspect of climate change impact on water balance components is the influence on crop productivity and yield. In the PRB, most of the agricultural area is limited near the downstream end (Figure 1–b). Projections indicate that in these sub-basins there is a notable decrease in precipitation in near future whereas a nominal decrease in mid and far future (Figure 8–a). The same trend is observed for ET, runoff, and groundwater discharge (Figure 8–b, 9–a, and 9–b). These changing trends raise concerns as reduced precipitation not only diminishes crop yield but also hampers crop growth. Additionally, runoff plays a crucial role in carrying nutrients from the soil and maintaining soil moisture. Consequently, a decrease in runoff can result in reduced nutrient availability for plant uptake and a decline in soil moisture content. The reduced flow of groundwater is also worrisome, as it can increase the risk of soil salinity and negatively impact soil fertility. However, it is essential to recognize that these changes are multifaceted, requiring further investigation into the associated agricultural risks within the region. Consequently, strategic planning and effective water resource management are imperative to address these challenges and mitigate their potential impacts on agriculture.”

COMMENT 2: Review:

It is a very difficult to discover the main findings and outputs of the study because the river basin is too big and the scale too course. It would a very useful to summarize all the results in the table, where the individual sub-basins will be compared, included their percentual changes of the water balance. Otherwise, the main findings are extinguished. Therefore, I recommend accepting the manuscript after minor revision.

Ans: We thank the Reviewer for this comment.

The results for all the sub-basins has been added in supplementary data and the information is mentioned in the text.

Refer Line No 419–421 and Supplementary Material.

Further, the results for all the above-mentioned water balance components, along with percolation and streamflow, averaged over the basin, are mentioned in Table 2 and the results for each sub-basins is mentioned in the Supplementary material (Table S1–S4).

COMMENT 3: Title

Although the title of the article has been slightly modified, it is still too general. In the title should be clearly specify which water balance is considering. Is it a water balance of a system, soil profile or a river basin? I recommend specifying in the title global climate change instead of the climate change. In addition, the term: “sustainable water future” is a very general: could you describe if it concerns the quality of quantity?

Ans: We thank the Reviewer for this suggestion. The title has been changed mentioning that the water balance is considered at river basin scale. Further, the study considers climate change impact at regional scale therefore we suppose that adding the term “Global” climate change may be misleading for readers.

In addition to this, the quantified assessment of climate change's impact on water balance components is a crucial aspect to be considered when planning and managing for a sustainable future. This objective has been explicitly stated in the study's aim within the introduction section, which helps the readers to understand the gist of the research content and also avoids a lengthy title.

Refer Line No. 2–4

“Assessing the Vulnerability of Water Balance to Climate Change at River Basin Scale in Humid Tropics: Implications for a Sustainable Water Future”

COMMENT 4: Abstract: The abstract exceeds the maximum amount of the words, see the journal sustainability requirements: the abstract should be a total of about 200 words maximum.

The text of the manuscript contains typos, e.g.:

Line 105: „is” instead of “ils”

Ans: Thank you for this comment.

The abstract size has been reduced and all other corrections has been made.

Refer Line no. 9–33 and 107.

COMMENT 5: Introduction

Would you be so kind to describe, what is the novelty of this study? This information is still missing within the text, and it must be included there.

Ans: Thank you for this comment.

This study discusses the impact of climate change on the water balance components at river basin scale in a humid tropical region based on the latest SSP climate change scenarios.  The Periyar river basin is considered in this study which is a complex watershed impaired with various anthropogenic constraints including controlled reservoir operation and restrictions on landuse changes. In addition, the unique climatic conditions due to orography influence makes it an important area for study. No past study has been done on this area to consider climate change impact on water balance with the latest SSP climate change scenarios. The details have been added in the text.

Refer Line no 135–141.

Thus, assessing the impact of future climate change, considering latest SSP scenarios, on water balance components within this complex watershed is the main focus of this study. This aspect has not been previously addressed in any other study in this region. The SWAT model is employed to conduct this assessment, allowing for a comprehensive evaluation of the effects. The datasets from three GCMs are utilized to analyze the change till 2100 under the SSP 245 and SSP 585 scenarios.

COMMENT 6: Materials and Methods/ Study area description and input data details/ Study area description

The study is describing the water balance of the river basin. However, the main and important information such as the hydrological water regime of the river basin and its descriptions is missing yet. That is crucial and necessary. When is the highest discharges in which period? What about the average annual temperature?

Ans: We thank the Reviewer for this suggestion.

The components of hydrological water regime including annual precipitation, evapotranspiration, surface runoff, and groundwater discharge for PRB are discussed in Section 3.3. (Refer Line No. 325–338)

A brief discussion on the water balance components is mentioned in the study area description. (Refer Line No. 153–157 and 169–172)

Since it is a large-elongated basin, the average annual temperature varies from 28Ëš C (near upstream) to 30Ëš C (near downstream end). This information has been added in the text.

Additional information on maximum daily flow is also added in the text.

Refer Line No. 155–156 and 170–172.

Since it is a large-elongated basin, the average annual temperature varies from 28Ëš C near upstream to 30Ëš C near the downstream end.”

“The maximum daily discharge measured between 1989 and 2017 peaked at 6324 m3/s. Notably, the last week of July or the first and second weeks of August experience the majority, or around 80% of the daily discharges over 2000 m3/s.”

COMMENT 7: Topographical and meteorological data

Could you describe why the specific data sets have been chosen? Do you have some special reason for that?

Ans: Thank you for this comment.

These datasets were chosen based on the requirement of the SWAT model and further on the basis of the availability and suitability of the data in the study area in reference to past literature. The same has been mentioned in the text.

Refer Line No. 230–233 and 216–217.

“SWAT input requires physical parameters, including soil type, LULC, DEM, and meteorological variables, including minimum and maximum temperature, solar radiation, precipitation, and wind speed.”

“These GCMs were selected on the basis of their ability to replicate the Indian Summer Monsoon [40] and performance in past studies [41].”

COMMENT 8: GCM climate data

Could you describe which scenario is the most favourable and what are the differences between the models?

Ans: We thank the Reviewer for this comment. The SSP 245 corresponds to the development pathways consistent with historical patterns and with medium radiating forcings (up to 4.5 W/m2) and SSP 585 scenario corresponds to high radiating forcings (up to 8.5 W/m2) till 2100. It has been mentioned in the text.

It is difficult to comment on the most favorable scenario as consequences of climate change vary with region and the effects are highly uncertain. However, for general comment, the SSP 585 proposes the most unfavorable climatic changes with high temperature rise and subsequent effects.

Refer Line No 219–227.

“In this study, two socioeconomic pathways: SSP 245 and SSP 585 were considered for analysis from each GCM. The SSP 245 corresponds to the development pathways consistent with historical patterns and with medium radiating forcings (up to 4.5 W/m2) in 2100. This scenario represents the medium range of the possible future. Whereas the SSP 585 corresponds to development pathways representing high industrial/economic growth and fossil fuel resources consumption with minimal efforts over reducing environmental concerns. This scenario corresponds to high radiating forcings (up to 8.5 W/m2) in 2100 [18] with potentially high challenges to mitigation strategies.

COMMENT 9: Results

Would you be so kind to summarize all the main findings in the more comprehensive way, please? Especially part “3.3. Impact of climate change on water balance components” is very unclear. Could you, please summarize all the parts of the water balance, please?

Ans: Thank you for this comment.

In Section 3.3, the results for the impact of climate change on the water balance components across the sub-basins have been discussed in a detail. The overall change averaged over the PRB has been tabulated in Table 2.

The discussion on the observed changes has been done in Section 3.4.

The addition of Supplementary data representing the change across all the sub-basins will help the readers to better understand the results in a comprehensive way.

These finding highlights the main result of the research, demonstrating the spatial variability of responses to climate change across the sub-basins in large river basin, despite their similar climatic conditions. This has been added in the conclusions.

Refer Line no. 419–421 and 544–546.

This finding highlights the main result of the research, demonstrating the spatial variability of responses to climate change across the sub-basins in a large river basin, despite their similar climatic conditions.

COMMENT 10: Concussions

How the results will be used? What was the main purpose of the study?

Ans: Thank you for this comment.

The main purpose of this study is to highlight the insights about the possible changes in the water balance components due to climate change at a river basin scale. This will help in identifying water scarce/ surplus regions and having this information would supplement to decision making in management processes. Furthermore, having the knowledge of these probable changes, specific strategies can be planned by government agencies, NGO’s, agricultural industry, and other stakeholders.

This would encourage for optimum water consumption practices, allowing for better management and allocation of water resources.

Refer Line No. 553–558.

This study also underscores the significance of using modeling approaches in developing long-term strategies for water resource management that can support sustainable development in the face of climate change. Some of these strategies may include flood control in areas where excess runoff is observed. While others may include utilizing this information on excess runoff water availability in future to enhance productivity by hydro-power generation, agricultural productivity, increase water recreation and tourism.

References

Neitsch, S. L., Arnold, J. G., Kiniry, J. R., & Williams, J. R. (2011). Soil and water assessment tool theoretical documentation version 2009. Texas Water Resources Institute.
